# DOZERFORMER: SEQUENCE ADAPTIVE SPARSE TRANSFORMER FOR MULTIVARIATE TIME SERIES FORECASTING

## ABSTRACT

Transformers have achieved remarkable performance in multivariate time series(MTS) forecasting due to their capability to capture long-term dependencies. However, the canonical attention mechanism has two key limitations: (1) its quadratic time complexity limits the sequence length, and (2) it generates future values from the entire historical sequence. To address this, we propose a Dozer Attention mechanism consisting of three sparse components: (1) Local, each query exclusively attends to keys within a localized window of neighboring time steps. (2) Stride, enables each query to attend to keys at predefined intervals. (3) Vary, allows queries to selectively attend to keys from a subset of the historical sequence. Notably, the size of this subset dynamically expands as forecasting horizons extend. Those three components are designed to capture essential attributes of MTS data, including locality, seasonality, and global temporal dependencies. Additionally, we present the Dozerformer Framework, incorporating the Dozer Attention mechanism for the MTS forecasting task. We evaluated the proposed Dozerformer framework with recent state-of-the-art methods on nine benchmark datasets and confirmed its superior performance. Code is available at github.com/Dozerformer.

## 1 INTRODUCTION

Multivariate time series (MTS) forecasting endeavors to adeptly model the dynamic evolution of multiple variables over time from their historical records, thereby facilitating the accurate prediction of future values. This task holds significant importance in various applications Petropoulos et al. (2022). The advent of deep learning has significantly advanced MTS forecasting, various methods are proposed based on Recurrent Neural Networks (RNN) Lai et al. (2018), Convolutional Neural Networks (CNN) Shih et al. (2019); Wang et al. (2023). In recent years, the Transformer has demonstrated remarkable efficacy in MTS forecasting. This notable performance can be attributed to its inherent capacity to effectively capture global temporal dependencies across the entirety of the historical sequence. The Transformer is initially proposed for natural language processing(NLP) Vaswani et al. (2017) tasks, with its primary objective of extracting information from words within sentences. It achieved great success in the NLP field and swiftly extended its impact in computer vision Dosovitskiy et al. (2021) and time series analysis Wen et al. (2023). The critical challenge for applying transformers in MTS forecasting is to modify the attention mechanism based on the characteristics of time series data. Informer Zhou et al. (2021), Autoformer Wu et al. (2021), FEDformer Zhou et al. (2022), and Crossformer Zhang & Yan (2023) modified the canonical full attention mechanism to improve its efficiency and achieved promising results in MTS forecasting. Dlinear Zeng et al. (2023) challenges the effectiveness of the transformer by proposing a straightforward linear layer to infer historical records to predictions directly and outperforming aforementioned Transformer-based methods on benchmark datasets.

Nevertheless, these methods generate predictions for all future time steps from the entire historical sequence. Thus, they ignored that the look-back window of historical time steps is critical in generating accurate predictions. For example, predicting the value at horizon 1 using the entire historical sequence is suboptimal and inefficient. Figure 1(b) shows the full attention mechanism generating predictions, it generates predictions for all $O$ future time steps from the entire historical sequence of length $I$. They also ignored the characteristics of MTS data, like locality and seasonality. Fig-

ure 1(a) shows the heatmap of correlation among 168 time steps of four datasets that indicate clear locality and seasonality. An attention mechanism should only utilize those time steps that have a high correlation with the prediction target.

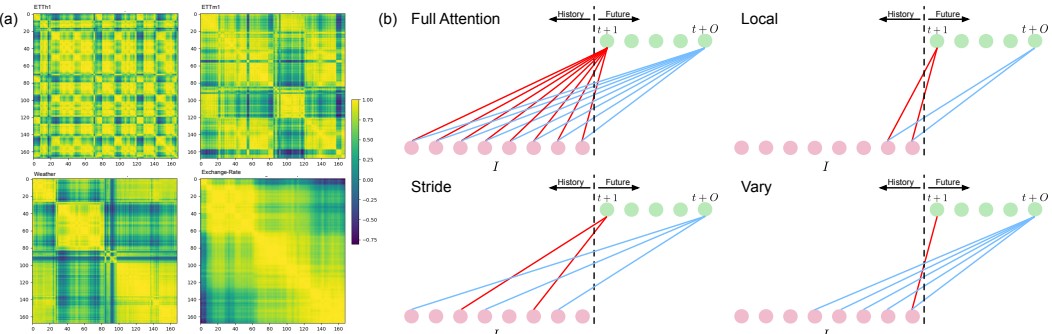

Figure 1: (a) The heatmap illustrates the correlation among 168 time steps for the ETTh1, ETTm1, Weather, and Exchange-Rate datasets. (b) Full Attention: Generates predictions using the entire historical sequence. Local Component: Utilizes time steps within a specified window. Stride Component: Utilizes time steps at fixed intervals from the target. Vary Component: Adaptively uses historical time steps as the forecasting horizon increases.

To address those issues, we propose Dozerformer, a novel framework that incorporates an innovative attention mechanism comprising three key components: Local, Stride, and Vary. Figure 1(b) illustrates the concept of the Local, Stride, and Vary components. They leverage historical time steps based on the distinctive characteristics of MTS data. The Local component exclusively considers time steps within a window determined by the target time step. The Stride component selectively employs time steps that are at fixed intervals from the target time step. Lastly, the Vary component dynamically adjusts its use of historical time steps according to the forecasting horizon—employing a shorter historical sequence for shorter horizons and a longer one for longer horizons. The contributions of our work are summarized as follows:

- We introduce a sequence-adaptive sparse Dozer Attention mechanism composed of three components: Local, Stride, and Vary. Each component aims to capture temporal dependencies from a select number of effective historical time steps. Notably, the Vary component dynamically expands its utilization of historical time steps as the forecasting horizons extend.

- We propose the Dozerformer framework for MTS forecasting, incorporating the Dozer Attention mechanism designed to model and predict the seasonal components within MTS data.

- The experimental results on nine benchmark datasets showcase Dozerformer's superior performance in terms of both accuracy and efficiency when compared to recent state-of-the-art methods.

## 2 RELATED WORK

### 2.1 MTS FORECASTING

MTS forecasting involves predicting values ($X_{pred} \in \mathbb{R}^{O \times D}$) of $D$ variables for future $O$ time steps, given their historical records within the look-back window ($X \in \mathbb{R}^{I \times D}$) spanning $I$ time steps. Deep learning has significantly advanced MTS forecasting. LSTNet Lai et al. (2018) employs CNN and RNN to capture both short-term and long-term temporal dependencies. TPA-LSTM Shih et al. (2019) introduces a temporal pattern attention mechanism to weight hidden states across historical time steps. Graph WaveNet Wu et al. (2019) pioneers the use of Graph Neural Networks (GNN) for MTS forecasting, jointly training an adjacency matrix to model spatial dependencies. MTGNN Wu et al. (2020) is the first GNN-based method for generic MTS forecasting. It utilizes GNN to model correlations among multiple variables and CNN with different filter sizes to model

multi-scale temporal dependencies. Transformers Wen et al. (2023) gained popularity in MTS forecasting due to their capacity to capture long-term temporal dependencies. The attention mechanism enables modeling temporal dependencies across all historical time steps. While CNN needs to be stacked, RNN needs to iteratively go through all time steps, and GNN is vulnerable to the adjacency matrix which indicates the correlation between variables. MICN Wang et al. (2023) replaces self-attention with a CNN-based local-global structure to model temporal dependencies across scales efficiently. Dlinear Zeng et al. (2023) challenges transformer effectiveness with a simple linear layer, achieving state-of-the-art performance. PatchTST Nie et al. (2023) highlights the significance of transformers in modeling temporal dependencies at sub-series resolution, independently of variables. In summary, transformers have pushed the boundaries of MTS forecasting.

## 2.2 SPARSE SELF ATTENTIONS

Transformers have achieved significant achievements in various domains, including NLP Vaswani et al. (2017), computer vision Dosovitskiy et al. (2021), and time series analysis Wen et al. (2023). However, their quadratic computational complexity limits input sequence length. Recent studies have tackled this issue by introducing modifications to the full attention mechanism. Longformer Beltagy et al. (2020) introduces a sparse attention mechanism, where each query is restricted to attending only to keys within a defined window or dilated window, except for global tokens, which interact with the entire sequence. Similarly, BigBird Zaheer et al. (2020) proposes a sparse attention mechanism consisting of Random, Local, and Global components. The Random component limits each query to attend a fixed number of randomly selected keys. The Local component allows each query to attend keys of nearby neighbors. The Global component selects a fixed number of input tokens to participate in the query-key production process for the entire sequence. In contrast to NLP, where input consists of word sequences, and computer vision Khan et al. (2022), where image patches are used, time series tasks involve historical records at multiple time steps. To effectively capture time series data's seasonality, having a sufficient historical record length is crucial. For instance, capturing weekly seasonality in MTS data sampled every 10 minutes necessitates approximately $6 \times 24 \times 7$ time steps. Consequently, applying the Transformer architecture to time series data is impeded by its quadratic computational complexity. To address this challenge, various methods have been proposed. Informer Zhou et al. (2021) introduces a ProbSparse attention mechanism, allowing each key to attend to a limited number of queries. This modification achieves $\mathcal{O}(I \log I)$ complexity in both computational and memory usage. Autoformer Wu et al. (2021) proposes an Auto-Correlation mechanism designed to model period-based temporal dependencies. It achieves $\mathcal{O}(I \log I)$ complexity by selecting only the top $logI$ query-key pairs. FEDformer Zhou et al. (2022) introduces frequency-enhanced blocks, including Fourier and Wavelet components, to transform queries, keys, and values into the frequency domain. The attention mechanism computes a fixed number of randomly selected frequency components from queries, keys, and values, resulting in linear complexity. Both Crossformer Zhang & Yan (2023) and PatchTST Nie et al. (2023) propose patching mechanisms that partition time series data into patches spanning multiple time steps, effectively reducing the total length of the historical sequence.

## 3 METHOD

In this section, we present the Dozerformer method, which incorporates the Dozer attention mechanism. Section 3.1 introduces the Dozerformer framework, which includes a transformer encoder-decoder pair and a linear layer designed for forecasting seasonal and trend components of MTS data, respectively. Section 3.2 provides a comprehensive description of the proposed Dozer attention mechanism, with its primary concept centered around the elimination of query-key pairs between input and output time steps that do not contribute to accuracy.

## 3.1 FRAMEWORK

The MTS forecasting task aims to infer values of $D$ variables for future $O$ time steps $X \in \mathbb{R}^{O \times D}$ from their historical records $X \in \mathbb{R}^{I \times D}$. Figure 2 illustrates the overall framework of Dozerformer. Initially, it decomposes the MTS data into seasonal and trend components, following previous methods Wu et al. (2021); Zhou et al. (2022); Zeng et al. (2023). Subsequently, the transformer and linear

models are employed to generate forecasts for the seasonal component $\mathbf{X}_s \in \mathbb{R}^{I \times D}$ and the trend component $\mathbf{X}_t \in \mathbb{R}^{I \times D}$, respectively.

The dimension invariant (DI) embed Zhang et al. (2023) transforms raw single-channel MTS data into multi-channel feature maps while preserving both the time step and variable dimensions. It further partitions the MTS data into multiple patches along the time step dimension, resulting in patched MTS embeddings denoted as $X_{enc} \in \mathbb{R}^{c \times N_{enc} \times p \times D}$. Here, $c$ represents the number of feature maps, $N_{enc} = \lceil I/p \rceil$ indicates the number of patches for the encoder, and $p$ represents the patch size. Similarly, the decoder's input, denoted as $X_{dec} \in \mathbb{R}^{c \times N_{dec} \times p \times D}$ (not shown in Figure 2), is derived from $L$ historical time steps and $O$ zero padding. In this context, $N_{dec} = \lceil (L+O)/p \rceil$ indicates the number of patches for the decoder. The transformer model is designed to capture temporal dependencies at multiple time step resolutions (sub-series level). While the transformer encoder and decoder adhere to the canonical transformer architecture Vaswani et al. (2017), they employ the proposed Dozer attention mechanism in place of the canonical full attention. A $1 \times 1$ CNN is employed to generate seasonal component predictions based on the learned latent representations from the transformer.

To predict the trend component, we employ a linear layer for the direct inference of trend forecasts from historical trend components. Subsequently, the forecasts for the seasonal and trend components are aggregated by summation to obtain the final predictions denoted as $\mathbf{X}_{pred} \in \mathbb{R}^{O \times D}$.

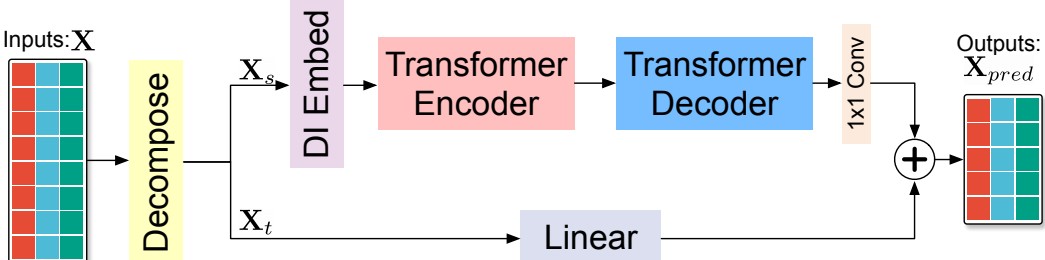

Figure 2: The architecture of our proposed Dozerformer framework.

## 3.2 DOZER ATTENTION

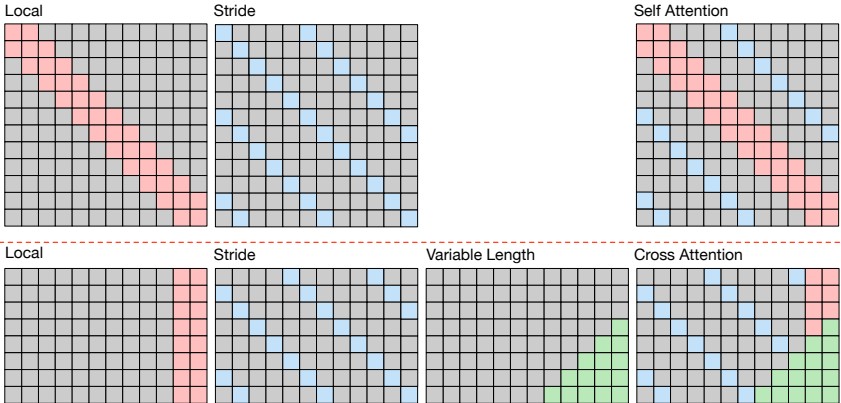

Figure 3: The illustration of the Dozer attention mechanism. **Upper:** The self-attention consists of Local and Stride. **Lower:** The cross attention consists of Local, Stride, and Vary.

The canonical scaled dot-product attention is as follows:

$$Q, K, V = \text{Linear}\left(\mathbf{X}_{enc}\right)$$
$$\text{Attention}\left(Q, K, V\right) = \text{Softmax}\left(QK^T / \sqrt{d_k}\right) V \tag{1}$$

where the $Q$, $K$, and $V$ represent the query, keys, and values obtained through embedding from the input sequence of the $d$-th series, denoted as $\mathbf{X}enc^d \in \mathbb{R}^{c \times Nenc \times p}$. It's noteworthy that we flatten both the feature map and patch size dimensions of $\mathbf{X}enc^d$, resulting in $\mathbf{X}enc^d \in \mathbb{R}^{N_{enc} \times (c \times p)}$, representing the latent representations of patches with a size of $p_k$. The canonical transformer exhibits two key limitations: firstly, its encoder's self-attention is afflicted by quadratic computational time complexity, thereby restricting the length of $N_{enc}$; secondly, its decoder's cross-attention employs the entire sequence to forecast all future time steps, entailing unnecessary computation for ineffective historical time steps. To overcome these issues, we propose a Dozer attention mechanism, comprising three components: Local, Stride, and Vary.

In Figure 3, we illustrate the sparse attention matrices of Local, Stride, and Vary components employed in the proposed Dozer attention mechanism. Squares shaded in gray signify zeros, indicating areas where the computation of the query-key product is redundant and thus omitted. These areas represent saved computations compared to full attention. The pink-colored squares correspond to the Local component, signifying that each query exclusively calculates the product with keys that fall within a specified window. Blue squares denote the Stride component, where each query selectively attends to keys that are positioned at fixed intervals. The green squares represent the Vary component, where queries dynamically adapt their attention to historical sequences of varying lengths based on the forecasting horizons. As a result, the Dozer attention mechanism exclusively computes the query-key pairs indicated by colored (pink, blue, and green) squares while efficiently eliminating the computation of gray query-key pairs.

### 3.2.1 LOCAL

The MTS data comprises variables that change continuously and exhibit locality, with nearby values in the time dimension displaying higher correlations than those at greater temporal distances. This characteristic is evident in Figure 3(a), which illustrates the strong locality within MTS data. Building upon this observation, we introduce the Local component, a key feature of our approach. In the Local component, each query selectively attends to keys that reside within a specified time window along the temporal dimension. Equation 2 defines the Local component as follows:

$$
\begin{aligned}
A_{i,j}^{self} &= \begin{cases} q_i * k_j, & \text{if } j \in \{|i - j| <= \lfloor w/2 \rfloor\} \\ 0, & \text{otherwise} \end{cases} \\
A_{i,j}^{cross} &= \begin{cases} q_i * k_j, & \text{if } j \in \{0 <= t - j <= \lfloor w/2 \rfloor\} \\ 0, & \text{otherwise} \end{cases}
\end{aligned}
\tag{2}
$$

In the case of self-attention, the Local component enables each query to selectively attend to a specific set of keys that fall within a defined time window. Specifically, each query in self-attention attends to $\lfloor w/2 \rfloor + 1$ keys, with $\lfloor w/2 \rfloor$ keys representing the neighboring time steps on each side of the query. For cross-attention, the concept of locality differs since future time steps cannot be utilized during forecasting. Instead, the locality is derived from the last $\lfloor w/2 \rfloor + 1$ time steps, situated at the end of the sequence. These steps encompass the most recent time points up to the moment of forecasting, denoted as time $t$.

### 3.2.2 STRIDE

Seasonality is a key characteristic of time series data, and to capture this attribute effectively, we introduce the Stride component. In this component, each query selectively attends to keys positioned at fixed intervals in the temporal dimension. To illustrate this concept, let's consider a query at time step $t$, denoted as $q_t$, and assume that the time series data exhibits a seasonality pattern with a period ($interval$). Equation 3 defines the Stride component as follows:

$$
A_{i,j} = \begin{cases} q_i * k_j & \text{if } j \in \{|i - j| \% interval == 0\} \\ 0, & \text{otherwise} \end{cases}
\tag{3}
$$

In self-attention, the Stride component initiates by computing the product between the query $q_t$ and the key at the same time step, denoted as $k_t$. It then progressively expands its attention to keys positioned at a temporal distance of $interval$ time steps from $t$, encompassing keys like $k_{t-interval}$, $k_{t+interval}$, and so forth, until it spans the entire sequence range. For cross-attention, the Stride component identifies time steps within the input sequence $X_I$ that are separated by multiples of

*interval* time steps, aligning with the seasonality pattern. Hence, the Stride component consistently computes attention scores (query-key products) using these selected keys from the Encoder's $I$ input time steps, yielding a total of $s = \lfloor I/interval \rfloor$ query-key pairs.

### 3.2.3 VARY

In the context of MTS forecasting scenarios employing the canonical attention mechanism, predictions for all future time steps are computed through a weighted sum encompassing the entirety of the historical sequence. Regardless of the forecasting horizon, this canonical cross-attention mechanism diligently computes all possible query-key pairs. However, it is important to recognize that harnessing information from the entire historical sequence does not necessarily improve forecasting accuracy, especially in the case of time series data characterized by evolving distributions, as exemplified by stock prices.

To tackle this challenge, we propose the Vary component, which dynamically expands the historical time steps considered as the forecasting horizon extends. We define the initial length of the historical sequence utilized as $v$. When forecasting a single time step into the future (horizon of 1), the Vary component exclusively utilizes $v$ time steps from the sequence. As the forecasting horizon gradually increases by 1, the history length used also increases by 1 from the starting vary window $v$. Equation 4 defines the Vary component as follows:

$$A_{i,j}^{cross} = \begin{cases} q_i * k_j, & \text{if } j \in \{t-1, t-2, \cdots, t-v+i-t\} \text{ and } i > t \\ 0, & \text{otherwise} \end{cases} \quad (4)$$

## 4 EXPERIMENTS

### 4.1 EXPERIMENTAL SETTINGS

**Datasets.** We conduct experiments on 9 public benchmarks Wu et al. (2021), including 4 ETT datasets(ETTh1, ETTh2, ETTm1, ETTm2), Traffic, Electricity, Weather, Exchange-Rate, and ILI. These are widely used real-world MTS datasets with different characteristics.

**Comparison methods.** We compare our Dozerformer with seven transformer-based methods (Informer Zhou et al. (2021), Autoformer Wu et al. (2021), FEDformer Zhou et al. (2022), Pyraformer Liu et al. (2022), Crossformer Zhang & Yan (2023), PatchTST Nie et al. (2023)), CNN-based method MICN Wang et al. (2023), and Linear method Dlinear Zeng et al. (2023).

**Evaluation metrics.** We utilize Mean Squared Error (MSE) and Mean Absolute Error (MAE) as the evaluation metrics to compare the forecasting accuracy.

**Training details.** We implemented the proposed Dozerformer in Pytorch and trained with Adam optimizer ($\beta_1 = 0.9$ and $\beta_2 = 0.99$). The initial learning rate is selected from $\{5e-5, 1e-4, 5e-4, 1e-3\}$ via grid search and updated using the Cosine Annealing scheme. The transformer encoder layer is set to 2 and the decoder layer is set to 1. We select the look-back window size of historical time steps utilized from $\{96, 192, 336, 720\}$ via grid search except for the ILI dataset which is set to 120. The patch size is selected from $\{24, 48, 96\}$ via grid search. The main results are repeated six times using seeds $1, 2022, 2023, 2024, 2025$, and $2026$. The ablations study and parameter sensitivity results are obtained from random seed 1.

### 4.2 MAIN RESULTS

Table 1 presents a quantitative comparison between the proposed Dozerformer and baseline methods across nine benchmarks using MSE and MAE, where the bolded and underlined case (dataset, horizon, and metric) represent the best and second-best results, respectively. The Dozerformer achieved superior performance than state-of-the-art baseline methods by performing the best in 48 cases and second-best in 22 cases. Compared to state-of-the-art baseline methods, Dozerformer achieved reductions in MSE: 0.4% lower than PatchTST Nie et al. (2023), 8.3% lower than Dlinear Zeng et al. (2023), 20.6% lower than MICN Wang et al. (2023), and a substantial 40.6% lower than Crossformer Zhang & Yan (2023). While PatchTST came closest to our method in performance, it utilizes full attention, making it less efficient in terms of computational complexity and memory usage. We

conclude that Dozerformer outperforms recent state-of-the-art baseline methods in terms of accuracy.

Table 1: Comparison of quantitative results (MSE&MAE) on nine datasets for forecasting horizons $O \in \{96, 192, 336, 720\}$ (For ILI, $O \in \{24, 36, 48, 60\}$). **Bold** and underlined highlight the best and second-best results, respectively.

| Methods | | Dozerformer | | PatchTST | | DLinear | | Crossformer | | MICN | | Pyraformer | | FEDformer | | Autoformer | | Informer | |
|---|---|---|---|---|---|---|---|---|---|---|---|---|---|---|---|---|---|---|---|
| Metric | | MSE | MAE | MSE | MAE | MSE | MAE | MSE | MAE | MSE | MAE | MSE | MAE | MSE | MAE | MSE | MAE | MSE | MAE |
| ETTh1 | 96 | **0.363** | **0.386** | 0.370 | 0.400 | 0.375 | 0.399 | 0.431 | 0.458 | 0.421 | 0.431 | 0.664 | 0.612 | 0.376 | 0.419 | 0.449 | 0.459 | 0.865 | 0.713 |
| | 192 | 0.405 | **0.413** | 0.413 | 0.429 | **0.405** | 0.416 | 0.420 | 0.448 | 0.474 | 0.487 | 0.790 | 0.681 | 0.420 | 0.448 | 0.500 | 0.482 | 1.008 | 0.792 |
| | 336 | 0.432 | **0.428** | **0.422** | 0.440 | 0.439 | 0.443 | 0.440 | 0.461 | 0.569 | 0.551 | 0.891 | 0.738 | 0.459 | 0.465 | 0.521 | 0.496 | 1.107 | 0.809 |
| | 720 | 0.453 | **0.459** | **0.447** | 0.468 | 0.472 | 0.490 | 0.519 | 0.524 | 0.770 | 0.672 | 0.963 | 0.782 | 0.506 | 0.507 | 0.514 | 0.512 | 1.181 | 0.865 |
| ETTh2 | 96 | **0.273** | **0.329** | 0.274 | 0.337 | 0.289 | 0.353 | 1.177 | 0.757 | 0.299 | 0.364 | 0.645 | 0.597 | 0.358 | 0.397 | 0.358 | 0.397 | 3.755 | 1.525 |
| | 168 | 0.341 | **0.374** | 0.341 | 0.382 | 0.383 | 0.418 | 1.206 | 0.796 | 0.441 | 0.454 | 0.788 | 0.683 | 0.429 | 0.439 | 0.456 | 0.452 | 5.602 | 1.931 |
| | 336 | 0.364 | 0.400 | **0.329** | **0.384** | 0.448 | 0.465 | 1.452 | 0.883 | 0.654 | 0.567 | 0.907 | 0.747 | 0.496 | 0.487 | 0.482 | 0.486 | 4.721 | 1.835 |
| | 720 | 0.394 | 0.432 | **0.379** | **0.422** | 0.605 | 0.551 | 2.040 | 1.121 | 0.956 | 0.716 | 0.963 | 0.783 | 0.463 | 0.474 | 0.515 | 0.511 | 3.647 | 1.625 |
| ETTm1 | 96 | **0.290** | **0.332** | 0.293 | 0.346 | 0.299 | 0.343 | 0.320 | 0.373 | 0.316 | 0.362 | 0.543 | 0.510 | 0.379 | 0.419 | 0.505 | 0.475 | 0.672 | 0.571 |
| | 192 | **0.328** | **0.356** | 0.333 | 0.370 | 0.335 | 0.365 | 0.400 | 0.432 | 0.363 | 0.390 | 0.557 | 0.537 | 0.426 | 0.441 | 0.553 | 0.496 | 0.795 | 0.669 |
| | 336 | **0.360** | **0.375** | 0.369 | 0.392 | 0.369 | 0.386 | 0.408 | 0.428 | 0.408 | 0.426 | 0.754 | 0.655 | 0.445 | 0.459 | 0.621 | 0.537 | 1.212 | 0.871 |
| | 720 | 0.416 | **0.410** | **0.416** | 0.420 | 0.425 | 0.421 | 0.582 | 0.537 | 0.481 | 0.476 | 0.908 | 0.724 | 0.543 | 0.490 | 0.671 | 0.561 | 1.166 | 0.823 |
| ETTm2 | 96 | **0.164** | **0.248** | 0.166 | 0.256 | 0.167 | 0.260 | 0.444 | 0.463 | 0.179 | 0.275 | 0.435 | 0.507 | 0.203 | 0.287 | 0.255 | 0.339 | 0.365 | 0.453 |
| | 192 | **0.217** | **0.285** | 0.223 | 0.296 | 0.224 | 0.303 | 0.833 | 0.657 | 0.307 | 0.376 | 0.730 | 0.673 | 0.269 | 0.328 | 0.281 | 0.340 | 0.533 | 0.563 |
| | 336 | **0.270** | **0.323** | 0.274 | 0.329 | 0.281 | 0.342 | 0.766 | 0.620 | 0.325 | 0.388 | 1.201 | 0.845 | 0.325 | 0.366 | 0.339 | 0.372 | 1.363 | 0.887 |
| | 720 | **0.354** | **0.378** | 0.362 | 0.385 | 0.397 | 0.421 | 0.959 | 0.752 | 0.502 | 0.490 | 3.625 | 1.451 | 0.421 | 0.415 | 0.433 | 0.432 | 3.379 | 1.338 |
| Traffic | 96 | 0.380 | **0.241** | 0.360 | 0.249 | 0.410 | 0.282 | 0.538 | 0.300 | 0.519 | 0.309 | 0.867 | 0.468 | 0.587 | 0.366 | 0.613 | 0.388 | 0.719 | 0.391 |
| | 192 | 0.386 | **0.244** | 0.379 | 0.256 | 0.423 | 0.287 | 0.515 | 0.288 | 0.537 | 0.315 | 0.869 | 0.467 | 0.604 | 0.373 | 0.616 | 0.382 | 0.696 | 0.379 |
| | 336 | 0.415 | **0.255** | 0.392 | 0.264 | 0.436 | 0.296 | 0.530 | 0.300 | 0.534 | 0.313 | 0.881 | 0.469 | 0.621 | 0.383 | 0.622 | 0.337 | 0.777 | 0.420 |
| | 720 | 0.471 | **0.281** | 0.432 | 0.286 | 0.466 | 0.315 | 0.573 | 0.313 | 0.577 | 0.325 | 0.896 | 0.473 | 0.626 | 0.382 | 0.660 | 0.408 | 0.864 | 0.472 |
| Electricity | 96 | **0.127** | **0.219** | 0.129 | 0.222 | 0.140 | 0.237 | 0.141 | 0.240 | 0.164 | 0.269 | 0.386 | 0.449 | 0.193 | 0.308 | 0.201 | 0.317 | 0.274 | 0.368 |
| | 192 | **0.142** | **0.234** | 0.147 | 0.240 | 0.153 | 0.249 | 0.166 | 0.265 | 0.177 | 0.177 | 0.378 | 0.443 | 0.201 | 0.315 | 0.222 | 0.334 | 0.296 | 0.386 |
| | 336 | **0.158** | **0.253** | 0.163 | 0.259 | 0.169 | 0.267 | 0.323 | 0.369 | 0.193 | 0.304 | 0.376 | 0.443 | 0.214 | 0.329 | 0.231 | 0.338 | 0.300 | 0.394 |
| | 720 | **0.196** | 0.291 | 0.197 | **0.290** | 0.203 | 0.301 | 0.404 | 0.423 | 0.212 | 0.321 | 0.376 | 0.445 | 0.246 | 0.355 | 0.254 | 0.361 | 0.373 | 0.439 |
| Weather | 96 | **0.145** | **0.190** | 0.149 | 0.198 | 0.176 | 0.237 | 0.158 | 0.231 | 0.161 | 0.229 | 0.622 | 0.556 | 0.217 | 0.296 | 0.266 | 0.336 | 0.300 | 0.384 |
| | 192 | **0.190** | **0.234** | 0.194 | 0.241 | 0.220 | 0.282 | 0.194 | 0.262 | 0.220 | 0.281 | 0.739 | 0.624 | 0.276 | 0.336 | 0.307 | 0.367 | 0.598 | 0.544 |
| | 336 | **0.237** | **0.271** | 0.245 | 0.282 | 0.265 | 0.319 | 0.495 | 0.515 | 0.278 | 0.331 | 1.004 | 0.753 | 0.339 | 0.380 | 0.359 | 0.395 | 0.578 | 0.523 |
| | 720 | **0.308** | **0.321** | 0.314 | 0.334 | 0.323 | 0.362 | 0.526 | 0.542 | 0.311 | 0.356 | 1.420 | 0.934 | 0.403 | 0.428 | 0.419 | 0.428 | 1.059 | 0.741 |
| Exchange | 96 | 0.086 | 0.210 | 0.896 | 0.209 | **0.081** | **0.203** | 0.323 | 0.425 | 0.102 | 0.235 | 1.748 | 1.105 | 0.148 | 0.278 | 0.197 | 0.323 | 0.847 | 0.752 |
| | 192 | 0.167 | 0.293 | 0.187 | 0.308 | **0.157** | **0.293** | 0.448 | 0.506 | 0.172 | 0.316 | 1.874 | 1.151 | 0.271 | 0.380 | 0.300 | 0.369 | 1.204 | 0.895 |
| | 336 | **0.266** | **0.380** | 0.349 | 0.432 | 0.305 | 0.414 | 0.840 | 0.718 | 0.272 | 0.407 | 1.943 | 1.172 | 0.460 | 0.500 | 0.509 | 0.524 | 1.672 | 1.036 |
| | 720 | **0.600** | **0.590** | 0.900 | 0.715 | 0.643 | 0.601 | 1.416 | 0.959 | 0.714 | 0.658 | 2.085 | 1.206 | 1.195 | 0.841 | 1.447 | 0.941 | 2.478 | 1.310 |
| ILI | 24 | 1.656 | 0.863 | **1.319** | **0.754** | 2.215 | 1.081 | 3.041 | 1.186 | 2.684 | 1.112 | 7.394 | 2.012 | 3.228 | 1.260 | 3.483 | 1.287 | 5.764 | 1.677 |
| | 36 | **1.491** | **0.832** | 1.579 | 0.870 | 1.963 | 0.963 | 3.406 | 1.232 | 2.667 | 1.068 | 7.551 | 2.031 | 2.679 | 1.080 | 3.103 | 1.148 | 4.755 | 1.467 |
| | 48 | 1.597 | 0.854 | **1.553** | **0.815** | 2.130 | 1.024 | 3.459 | 1.221 | 2.558 | 1.052 | 7.662 | 2.057 | 2.622 | 1.078 | 2.669 | 1.085 | 4.763 | 1.469 |
| | 60 | 1.770 | 0.887 | **1.470** | **0.788** | 2.368 | 1.096 | 3.640 | 1.305 | 2.747 | 1.110 | 7.931 | 2.100 | 2.857 | 1.157 | 2.770 | 1.125 | 5.264 | 1.564 |

## 4.3 COMPUTATIONAL EFFICIENCY

We analyze the computational complexity of transformer-based methods and present it in Table 2. For self-attention, the proposed Dozer self-attention achieved linear computational complexity w.r.t $I$ with a coefficient $(w + s)/p$. $w$ and $s$ denote the keys that the query attends with respect to Local and Stride components, which are small numbers (e.g. $w \in \{1, 3\}$ and $s \in \{2, 3\}$). In contrast, $p$ corresponds to the size of time series patches and is set to a larger value (e.g., $p \in 24, 48, 96$). We conclude that the coefficient $(w + s)/p$ is consistently smaller than 1 under the experimental conditions, highlighting the superior computational complexity performance of Dozer self-attention.

To analyze the computational complexity of cross-attention, it's essential to consider the specific design of the decoder for each method and analyze them individually. The Transformer calculates production between all $L + O$ queries and $I$ keys, with its complexity influenced by inputs for both the encoder and decoder. The Informer selects $log(L + O)$ queries and $I/2$ keys (time steps near $t$), resulting in $\mathcal{O}(log(L + O)I/2)$. Autoformer and FEDformer pad zeros to keys when $I$ is smaller than $L + O$ and only select the last $L + O$ keys when $I$ is greater than $L + O$. As a result, their cross-attention complexity is linear w.r.t $L + O$. The Crossformer decoder takes only $O$ zero paddings as input. Consequently, its cross-attention attends to $O/p$ queries and $I/p$ keys. It's worth noting that Crossformer also applies full attention across the variable dimension, so its complexity is additionally influenced by the variable count $D$. PatchTST solely employs the transformer encoder, making cross-attention irrelevant in this context.

The Local and Stride components of Dozer cross-attention specifically address $(L + O)/p$ queries, calculating the product of each query with $w$ keys within the local window and $s$ keys positioned at fixed intervals from the query. Consequently, their computational complexity is linear with respect to $L + O$, characterized by the coefficient $(w + s)/p$. The Vary component's complexity exhibits a quadratic relationship with respect to $O$, with a coefficient of $1/(2p^2)$. This quadratic complexity is notably more efficient compared to linear complexity when $O < 2p^2$ (e.g. 1152 when $p = 24$). Additionally, the Vary component maintains linear complexity concerning $O$ when $v$ is greater than 1, although its coefficient $(v - 1)/p$ should be significantly less than 1 for practical efficiency. Throughout all experiments, we consistently set $v$ to 1, rendering $(v - 1)O/p$ negligible.

The memory complexity of the Dozer Attention is $\mathcal{O}((w + s)I/p)$ for self-attention and $\mathcal{O}((w + s)(L + O)/p + (O/p)^2/2 + (v - 1)O/p)$ for cross-attention. It's worth noting that the analysis of Dozer Attention's computational complexity is equally applicable to its memory complexity. In conclusion, the Local and Stride components excel in efficiency by attending queries to a limited number of keys. As for the Vary component, it proves to be more efficient than linear complexity in practical scenarios, with its complexity only influenced by forecasting horizons.

Table 2: Computational complexity of self-attention and cross-attention. The Encoder's input historical sequence length is denoted $I$, and The decoder's input historical sequence length and forecasting horizons are denoted as $L$ and $O$, respectively. $D$ indicates the variable dimension of MTS data. $p$ is the patch size.

| Methods | Self-Attention | Cross-Attention |
|---|---|---|
| Transformer Vaswani et al. (2017) | $\mathcal{O}(I^2)$ | $\mathcal{O}((L + O)I)$ |
| Informer Zhou et al. (2021) | $\mathcal{O}(IlogI)$ | $\mathcal{O}(log(L + O)I/2)$ |
| Autoformer Wu et al. (2021) | $\mathcal{O}(IlogI)$ | $\mathcal{O}((L + O)log(L + O))$ |
| Fedformer Zhou et al. (2022) | $\mathcal{O}(I)$ | $\mathcal{O}(L + O)$ |
| Crossformer Zhang & Yan (2023) | $\mathcal{O}(D(I/p)^2)$ | $\mathcal{O}(DOI/p^2)$ |
| PatchTST Nie et al. (2023) | $\mathcal{O}((I/p)^2)$ | NA |
| Dozerformer | $\mathcal{O}((w + s)I/p)$ | $\mathcal{O}((w + s)(L + O)/p + (O/p)^2/2 + (v - 1)O/p)$ |

## 4.4 Ablation Studies

### 4.4.1 Local, Stride, and Vary Components Ablation

To investigate the impact of the Local, Stride, and Vary components individually, we conducted experiments by using only one of these components in the Dozer Attention. The results are summarized in Table 3, focusing on forecasting horizons of 96, 192, 336, and 720. The hyper-parameter of each component follows the setting of the main results. Overall, each component working alone performed slightly worse than the Dozer Attention, showing the effectiveness of each component in capturing the essential attributes of MTS data.

### 4.4.2 Attention Mechanism Comparison

To demonstrate the effectiveness of the proposed Dozer attention mechanism, we conducted a comparative analysis by replacing it with other attention mechanisms in the Dozerformer framework. Specifically, we compared it with canonical full attention Vaswani et al. (2017), Auto-Correlation Wu et al. (2021), Frequency Enhanced Auto-Correlation Zhou et al. (2022), and Prob-Sparse attention Zhou et al. (2021). The results of this comparison are presented in Table 4. The Dozer Attention outperformed other attention mechanisms, securing the top position with 22 best-case scenarios out of 32. Canonical full at-

Table 3: Ablation study of three key components of Dozer attention: Local, Stride, and Vary.

| Methods | Dozerformer | | Local | | Stride | | Vary | |
|---|---|---|---|---|---|---|---|---|
| Metric | MSE | MAE | MSE | MAE | MSE | MAE | MSE | MAE |
| ETTh$_1$ | 0.411 | **0.421** | 0.411 | 0.422 | 0.412 | 0.423 | **0.409** | 0.423 |
| ETTm$_1$ | 0.349 | 0.370 | **0.346** | **0.368** | 0.349 | 0.370 | 0.348 | 0.370 |
| Electricity | **0.156** | **0.249** | 0.161 | 0.255 | 0.157 | 0.251 | 0.162 | 0.256 |
| Weather | **0.220** | **0.255** | 0.223 | 0.257 | 0.228 | 0.260 | 0.226 | 0.258 |

tention performed as the second-best, achieving the best results in 9 cases. Significantly, the Dozer Attention reduced the average number of query-key pairs to 31.7% and 28.4% for self-attention in the encoder and cross-attention in the decoder, respectively, compared to full attention. This reduction represents a substantial decrease in computational complexity by eliminating the majority of

query-key pairs. The Auto-Correlation performed notably worse than the proposed Dozer Attention, with its MSE and MAE being 20.1% and 13.6% higher, respectively. The Frequency Enhanced Auto-Correlation exhibited performance that was 13.8% worse in MSE and 8% worse in MAE compared to the Dozer Attention. The ProbSparse Attention, while still competitive, performed slightly worse, with 5.7% higher MSE and 2% higher MAE than the Dozer Attention. We conclude that the proposed Dozer attention is just as capable as the canonical full attention but significantly more efficient and powerful than recent state-of-the-art attention mechanisms.

Table 4: Comparison of Dozerformer incorporating state-of-the-art attention mechanisms.

| Methods | QK Rat. | | Dozer | | Canonical | | AutoCorr | | FedAttn | | ProbSparse | |
|---|---|---|---|---|---|---|---|---|---|---|---|---|
| Metric | Self | Cross | MSE | MAE | MSE | MAE | MSE | MAE | MSE | MAE | MSE | MAE |
| ETTh₁ 96 | 19.1% | 17.9% | **0.362** | **0.386** | 0.364 | 0.386 | 0.376 | 0.399 | 0.380 | 0.396 | 0.368 | 0.393 |
| ETTh₁ 192 | 19.1% | 17.2% | **0.405** | **0.412** | 0.408 | **0.412** | 0.427 | 0.422 | 0.411 | 0.420 | 0.411 | 0.418 |
| ETTh₁ 336 | 19.1% | 16.1% | **0.432** | **0.428** | 0.441 | 0.429 | 0.460 | 0.460 | 0.484 | 0.450 | 0.440 | 0.434 |
| ETTh₁ 720 | 19.1% | 15.7% | 0.447 | 0.459 | 0.453 | **0.454** | 0.453 | 0.471 | 0.506 | 0.502 | **0.439** | 0.455 |
| ETTm₁ 96 | 32.4% | 26.6% | 0.292 | 0.334 | **0.284** | **0.327** | 0.330 | 0.367 | 0.297 | 0.338 | 0.292 | 0.334 |
| ETTm₁ 192 | 32.4% | 26.6% | 0.329 | 0.358 | 0.330 | **0.354** | 0.354 | 0.379 | 0.338 | 0.363 | **0.328** | 0.357 |
| ETTm₁ 336 | 32.4% | 25.9% | **0.359** | **0.376** | 0.369 | 0.378 | 0.374 | 0.392 | 0.367 | 0.382 | 0.362 | 0.379 |
| ETTm₁ 720 | 32.4% | 26.4% | **0.415** | **0.410** | 0.430 | 0.412 | 0.498 | 0.469 | 0.426 | 0.415 | 0.421 | 0.414 |
| Electricity 96 | 34.3% | 37.5% | **0.125** | **0.216** | 0.126 | 0.217 | 0.134 | 0.230 | 0.132 | 0.226 | 0.133 | 0.224 |
| Electricity 192 | 34.3% | 41.6% | 0.142 | 0.235 | **0.141** | **0.233** | 0.223 | 0.310 | 0.151 | 0.246 | 0.142 | 0.234 |
| Electricity 336 | 34.3% | 42.5% | **0.157** | **0.254** | 0.159 | 0.255 | 0.212 | 0.314 | 0.169 | 0.266 | 0.161 | 0.257 |
| Electricity 720 | 34.3% | 40.2% | **0.199** | **0.293** | 0.204 | 0.297 | 0.346 | 0.420 | 0.403 | 0.410 | 0.322 | 0.332 |
| Weather 96 | 37.7% | 31.6% | **0.147** | **0.193** | 0.151 | 0.196 | 0.179 | 0.230 | 0.152 | 0.199 | 0.151 | 0.195 |
| Weather 192 | 37.7% | 31.1% | 0.189 | 0.232 | **0.188** | **0.230** | 0.209 | 0.252 | 0.200 | 0.246 | 0.194 | 0.236 |
| Weather 336 | 37.7% | 30.3% | **0.237** | **0.272** | 0.239 | 0.274 | 0.321 | 0.332 | 0.279 | 0.309 | 0.249 | 0.280 |
| Weather 720 | 51.0% | 27.7% | **0.308** | **0.322** | 0.313 | 0.325 | 0.370 | 0.366 | 0.403 | 0.410 | 0.322 | 0.332 |

## 4.5 PARAMETER SENSITIVITY

### 4.5.1 EFFECT OF LOCAL, STRIDE, AND VARY SIZE

To investigate the effect of local size $w$, stride size $s$, and vary size $v$ to MTS forecasting accuracy, we conduct experiments and present the results in Figure 6. From the computational complexity aspect, smaller $w$, $s$, and $v$ indicate less computation. By increasing $w$, $s$, and $v$, the Dozer attention gets less sparse. However, including more query-key pairs doesn't improve the forecasting accuracy. Indicating the importance of removing less correlated historical time steps.

### 4.5.2 EFFECT OF LOOK-BACK WINDOW SIZE

The size of the look-back window, denoted as $I$, impacts primarily the stride component of the Dozerformer. Figure 7 illustrates the effect of look-back window size on seven methods at horizons 96 and 720. For the 96-hour horizon, no method benefits from a larger look-back window size except for the Exchange-Rate dataset. The Dozerformer consistently outperforms all baseline methods, indicating that time steps distant from the prediction target are less correlated and can even degrade forecasting accuracy. However, for the 720-hour horizon, a longer look-back window size enhances the Dozerformer's forecasting accuracy. The Exchange-Rate dataset, known for its locality, performs optimally with a small look-back window size for all methods. In conclusion, the optimal input sequence length varies with the forecasting horizon; shorter horizons favor a shorter look-back window, and vice versa.

## 5 CONCLUSIONS

This paper introduced a sequence adaptive sparse transformer framework named Dozerformer for MTS forecasting. The Dozerformer incorporates a sparse Dozer attention mechanism which consists of local, stride, and vary components to selectively attend each query to keys based on the characteristic of MTS data and forecasting horizons. We demonstrated the superior performance of Dozerformer against recent state-of-the-art methods in both accuracy and efficiency. For future work, we plan to further modify the Dozer attention to model multi-scale temporal dependencies.

AUTHOR CONTRIBUTIONS

If you'd like to, you may include a section for author contributions as is done in many journals. This is optional and at the discretion of the authors.

ACKNOWLEDGMENTS

Use unnumbered third level headings for the acknowledgments. All acknowledgments, including those to funding agencies, go at the end of the paper.

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

# A APPENDIX

## A.1 DATASETS DETAILS

Table 5 shows the characteristics of benchmark datasets. The details of nine benchmark datasets utilized for the evaluation of methods are summarized as follows:

- ETT[1]: The ETT (Electricity Transformer Temperature) serves as a vital indicator for long-term electric power infrastructure deployment. Informer Zhou et al. (2021) gathered data on electricity transformers, encompassing seven key indicators such as oil temperature and useful load, from two counties in China. Among these, ETTh1 and ETTh2 datasets consist of 17,420 samples, recorded at hourly intervals. On the other hand, ETTm1 and ETTm2 datasets encompass 69,680 samples, with measurements taken every 15 minutes.

- Traffic[2]: This dataset sourced from the California Department of Transportation, comprises road occupancy rate measurements, ranging from 0 to 1, gathered from 862 freeway sites within the San Francisco Bay area. This publicly available data spans over a decade. Lai **?** meticulously collected hourly data over a period of 48 months (2015-2016), resulting in a total of 17,544 data samples.

- Electricity[3]: This dataset, available from the UCI Machine Learning Repository, comprises the electricity consumption of 321 clients, measured in kWh at hourly intervals from 2012 to 2014, resulting in a total of 26,304 data samples.

- Weather[4]: This dataset encompasses 21 meteorological indicators, with recordings captured at 10-minute intervals over the span of a year in Germany, resulting in a total of 52,696 samples.

- Exchange-Rate[5]: This dataset spans a period of 27 years, from 1990 to 2016, and includes daily exchange rates for eight major world economies: Australia, Britain, Canada, Switzerland, China, Japan, New Zealand, and Singapore. In total, there are 7,588 data samples available.

- ILI[6]: This dataset comprises patient data with seven indicators spanning from 2002 to 2021, with weekly sampling, resulting in a total of 966 samples. Notably, this dataset stands out due to its unique forecasting horizon setting.

Table 5: Statistics of the benchmarking datasets

| Dataset | Series | Unit | Size | Length |
|---------|--------|------|------|--------|
| ETTh1 | 7 | 1 hour | 17,420 | 2 years |
| ETTh2 | 7 | 1 hour | 17,420 | 2 years |
| ETTm1 | 7 | 15 minutes | 69,680 | 2 years |
| ETTm2 | 7 | 15 minutes | 69,680 | 2 years |
| Traffic | 862 | 1 hour | 17,544 | 24 months |
| Electricity | 321 | 1 hour | 26,304 | 36 months |
| Weather | 21 | 10 minutes | 52,696 | 1 year |
| Exchange-Rate | 8 | 1 day | 7,588 | 27 years |
| ILI | 7 | 1 week | 966 | 20 years |

---

[1] https://github.com/zhouhaoyi/ETDataset
[2] http://pems.dot.ca.gov
[3] https://archive.ics.uci.edu/ml/datasets/ElectricityLoadDiagrams20112014
[4] https://www.bgc-jena.mpg.de/wetter/
[5] https://github.com/laiguokun/multivariate-time-series-data
[6] https://gis.cdc.gov/grasp/fluview/fluportaldashboard.html

## A.2 Detailed Computation of the Dozerformer

Given MTS input $X \in \mathbb{R}^{I \times D}$, the first step is to decompose it into seasonal and trend components Wu et al. (2021); Zhou et al. (2022); Zeng et al. (2023) as follows:

$$\mathbf{X}_t = \text{mean}\left(\sum_{i=1}^{n} \text{AvgPool}\left(\text{Padding}\left(\mathbf{X}\right)\right)_i\right)$$

$$\mathbf{X}_s = \mathbf{X} - \mathbf{X}_t \tag{5}$$

Where $\mathbf{X}_s \in \mathbb{R}^{I \times D}$ and $\mathbf{X}_t \in \mathbb{R}^{I \times D}$ are seasonal and trend-cyclical components, respectively.

A simple linear layer is utilized to model the trend component and generate trend component predictions.

$$\mathbf{X}_t^{pred} = \text{Linear}\left(\mathbf{X}_t\right) \tag{6}$$

where $\mathbf{X}_t^{pred} \in \mathbb{R}^{O \times D}$ is the prediction for trend component. The $\text{Linear}$ operation directly generates $O$ future trend values by projecting from $I$ historical trend values.

The transformer encoder-decoder pair utilizing the Dozer attention is employed as the seasonal model. The DI embedding Zhang et al. (2023) is utilized to partition time series data into patches and embed those patches into tokens as the input for the transformer.

$$\mathbf{X}_{emb} = \text{Conv}\left(\mathbf{X}_s\right)$$

$$\mathbf{X}_{pat} = \text{Patch}\left(\mathbf{X}_{emb}, \mathbf{X_0}, p\right) \tag{7}$$

where the $\mathbf{X}_s \in \mathbb{R}^{1 \times I \times D}$ is the seasonal components of input, $\mathbf{X}_{emb} \in \mathbb{R}^{c \times I \times D}$ represents $c$ feature maps embedded by a convolutional layer with kernel size of $3 \times 1$. The $\text{Patch}$ procedure divides the time series into $N = \lceil I/p \rceil$ non-overlapping patches of size $p$, yielding $\mathbf{X}_{pat} \in \mathbb{R}^{c \times N \times p \times D}$. The $\mathbf{X_0}$ is zero-padding when the input sequence length $I$ is not divisible by the patch size $p$.

The transformer encoder and decoder employ the Dozer attention as follows:

$$Q, K, V = \text{Linear}\left(\mathbf{X}_{pat}^d\right)$$

$$\text{Attention}\left(Q, K, V\right) = \text{Softmax}\left(Q\overline{K}^T / \sqrt{d_k}\right) V \tag{8}$$

where the $\overline{K}$ is the subset of keys selected by the characteristics of datasets. Note that the $\mathbf{X}_{pat}^d$ is one variable of the MTS data, following the channel-independent design of patchTST Nie et al. (2023). We follow the canonical multi-head attention utilizing the attention mechanism presented in Equation 8 as follows:

$$Q_h, K_h, V_h = \text{Linear}\left(Q, K, V\right)_h$$

$$\mathbf{H}_h^d = \text{Attention}\left(Q_h, K_h, V_h\right) \tag{9}$$

$$\mathbf{H}^d = \text{Linear}\left(\text{Concat}\left(\mathbf{H}_1^k, \cdots, \mathbf{H}_h, \cdots\right)\right)$$

Then, a $\text{Conv}$ layer with a kernel size of $1 \times 1$ is utilized to reduce the number of feature maps from $c$ to 1 as follows.

$$\mathbf{X}_s^{pred} = \text{Conv}\left(\mathbf{H}\right) \tag{10}$$

where $\mathbf{X}_s^{pred}$ is the predictions for the seasonal components. Lastly, the seasonal component and trend component are summed elementwisely as follows to generate final predictions.

$$\mathbf{X}_{pred} = \mathbf{X}_s^{pred} + \mathbf{X}_t^{pred} \tag{11}$$

where $\mathbf{X}_{pred} \in \mathbb{R}^{O \times D}$ is the predictions for the future $O$ time steps of the $D$ variables in the MTS data.

## A.3 More Experimental Results

### A.3.1 The quantitative results in MASE

Table 6 illustrates the quantitative comparison results between the proposed Dozerformer and baseline methods using the Mean Absolute Scaled Error (MASE) metric.

Table 6: Comparison of quantitative results (MASE) on nine datasets for forecasting horizons $O \in \{96, 192, 336, 720\}$ (For ILI, $O \in \{24, 36, 48, 60\}$). **Bold** and underlined highlight the best and second-best results, respectively.

| Methods | | Dozerformer | PatchTST | DLinear | Crossformer | MICN | Pyraformer | FEDformer | Autoformer | Informer |
|---|---|---|---|---|---|---|---|---|---|---|
| Metric | | MASE | MASE | MASE | MASE | MASE | MASE | MASE | MASE | MASE |
| ETTh₁ | 96 | 0.542 | 0.561 | 0.559 | 0.604 | 0.563 | 0.642 | 0.587 | 0.858 | 1 |
| | 192 | 0.564 | 0.585 | 0.567 | 0.664 | 0.585 | 0.611 | 0.611 | 0.929 | 1.08 |
| | 336 | 0.576 | 0.591 | 0.595 | 0.740 | 0.630 | 0.619 | 0.625 | 0.991 | 1.08 |
| | 720 | 0.608 | 0.619 | 0.648 | 0.888 | 0.661 | 0.693 | 0.670 | 1.034 | 1.14 |
| ETTh₂ | 96 | 0.780 | 0.798 | 0.836 | 0.862 | 0.886 | 1.795 | 0.940 | 1.414 | 3.613 |
| | 168 | 0.792 | 0.807 | 0.883 | 0.959 | 0.875 | 1.683 | 0.928 | 1.443 | 4.082 |
| | 336 | 0.788 | 0.755 | 0.915 | 1.116 | 0.889 | 1.739 | 0.958 | 1.470 | 3.612 |
| | 720 | 0.835 | 0.816 | 1.065 | 1.384 | 0.905 | 2.169 | 0.916 | 1.514 | 3.143 |
| ETTm₁ | 96 | 0.500 | 0.520 | 0.515 | 0.544 | 0.563 | 0.560 | 0.630 | 0.766 | 0.858 |
| | 192 | 0.517 | 0.536 | 0.528 | 0.565 | 0.560 | 0.626 | 0.639 | 0.778 | 0.969 |
| | 336 | 0.531 | 0.554 | 0.545 | 0.602 | 0.581 | 0.606 | 0.649 | 0.926 | 1.231 |
| | 720 | 0.562 | 0.576 | 0.577 | 0.652 | 0.617 | 0.737 | 0.672 | 0.993 | 1.128 |
| ETTm₂ | 96 | 0.758 | 0.780 | 0.792 | 0.838 | 0.814 | 1.411 | 0.875 | 1.545 | 1.381 |
| | 192 | 0.770 | 0.797 | 0.816 | 1.013 | 0.832 | 1.771 | 0.884 | 1.814 | 1.517 |
| | 336 | 0.788 | 0.802 | 0.834 | 0.946 | 0.856 | 1.513 | 0.892 | 2.060 | 2.163 |
| | 720 | 0.813 | 0.827 | 0.905 | 1.053 | 0.866 | 1.618 | 0.892 | 3.120 | 2.877 |
| Traffic | 96 | 0.223 | 0.230 | 0.261 | 0.286 | 0.297 | 0.278 | 0.339 | 0.433 | 0.362 |
| | 192 | 0.224 | 0.235 | 0.264 | 0.289 | 0.309 | 0.265 | 0.343 | 0.429 | 0.348 |
| | 336 | 0.233 | 0.241 | 0.270 | 0.285 | 0.306 | 0.273 | 0.349 | 0.428 | 0.383 |
| | 720 | 0.256 | 0.260 | 0.287 | 0.296 | 0.319 | 0.285 | 0.348 | 0.431 | 0.430 |
| Electricity | 96 | 0.232 | 0.234 | 0.250 | 0.284 | 0.287 | 0.254 | 0.325 | 0.474 | 0.389 |
| | 192 | 0.247 | 0.252 | 0.262 | 0.300 | 0.304 | 0.279 | 0.331 | 0.466 | 0.406 |
| | 336 | 0.264 | 0.269 | 0.277 | 0.316 | 0.312 | 0.383 | 0.342 | 0.460 | 0.409 |
| | 720 | 0.299 | 0.297 | 0.308 | 0.329 | 0.328 | 0.433 | 0.364 | 0.456 | 0.450 |
| Weather | 96 | 0.749 | 0.779 | 0.933 | 0.901 | 0.866 | 0.912 | 1.165 | 2.188 | 1.511 |
| | 192 | 0.801 | 0.825 | 0.965 | 0.962 | 0.893 | 0.899 | 1.150 | 2.136 | 1.863 |
| | 336 | 0.802 | 0.834 | 0.943 | 0.979 | 0.905 | 1.523 | 1.124 | 2.227 | 1.547 |
| | 720 | 0.816 | 0.847 | 0.918 | 0.903 | 0.911 | 1.375 | 1.086 | 2.370 | 1.880 |
| Exchange | 96 | 1.072 | 1.068 | 1.035 | 1.198 | 1.193 | 2.169 | 1.418 | 5.637 | 3.836 |
| | 192 | 1.016 | 1.068 | 1.013 | 1.093 | 1.190 | 1.753 | 1.314 | 3.982 | 3.096 |
| | 336 | 0.961 | 1.090 | 1.045 | 1.027 | 1.131 | 1.813 | 1.262 | 2.959 | 2.616 |
| | 720 | 0.867 | 1.043 | 0.882 | 0.966 | 1.095 | 1.408 | 1.234 | 1.770 | 1.923 |
| ILI | 24 | 0.507 | 0.443 | 0.635 | 0.653 | 0.549 | 0.697 | 0.740 | 1.182 | 0.985 |
| | 36 | 0.441 | 0.461 | 0.511 | 0.566 | 0.488 | 0.653 | 0.573 | 1.078 | 0.778 |
| | 48 | 0.475 | 0.453 | 0.569 | 0.585 | 0.522 | 0.679 | 0.599 | 1.144 | 0.817 |
| | 60 | 0.529 | 0.469 | 0.653 | 0.661 | 0.553 | 0.778 | 0.689 | 1.252 | 0.932 |

### A.3.2 THE MODEL COMPLEXITY OF ALL METHODS

To assess the efficiency of the proposed Dozerformer in comparison with baseline methods, we selected ETTh1 as an illustrative example and evaluated the Params (total number of learnable parameters), FLOPs (floating-point operations), and Memory (maximum GPU memory consumption). With a historical length set at 720 and a forecasting horizon of 96, Table 7 presents the quantitative results of this efficiency comparison. Dlinear Zeng et al. (2023), characterized by a straightforward architecture employing a linear layer to directly generate forecasting values from historical records, achieved the best efficiency among the methods considered. Notably, the proposed Dozerformer exhibited superior efficiency compared to other baseline methods based on CNN or transformers. It is essential to highlight that patchTST Nie et al. (2023) is the only method that did not significantly lag behind Dozerformer in accuracy. However, its parameter count is six times greater than that of Dozerformer, and FLOPs are 17 times larger.

Table 7: The Model complexity results in validation dataset for input length of 720 and forecasting horizon 96

| Models | Dozerformer | PatchTST | DLinear | Crossformer | MICN | Pyraformer | FEDformer | Autoformer | Informer |
|---|---|---|---|---|---|---|---|---|---|
| Params (M) | 1.61 | 10.73 | 0.13 | 52.99 | 18.78 | 42.36 | 215.01 | 9.05 | 11.32 |
| FLOPs(G) | 14.91 | 256.39 | 0.06 | 1046.19 | 1168.99 | 383.32 | 790.37 | 142.98 | 312.5 |
| Memory (M) | 103.59 | 611.68 | 4.09 | 2146.74 | 775.88 | 550.69 | 1814.01 | 315.44 | 635.38 |

### A.3.3 VISUALIZATION OF FORECASTING RESULTS

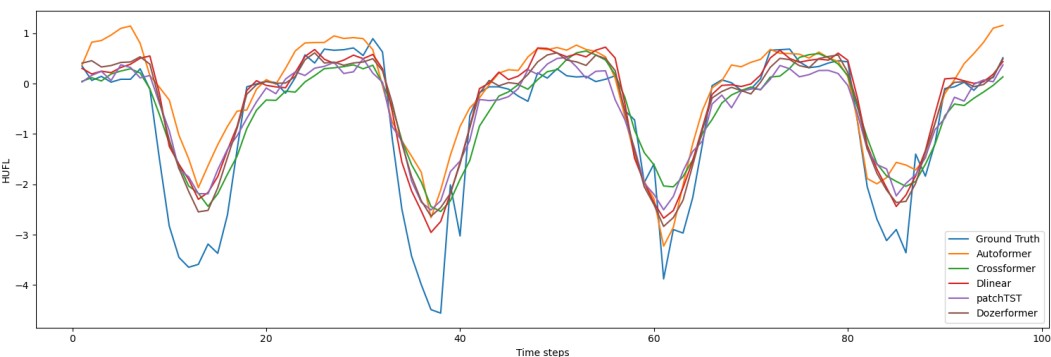

Figure 4: Visualizations of forecasting results on ETTh1 dataset at horizon 96.

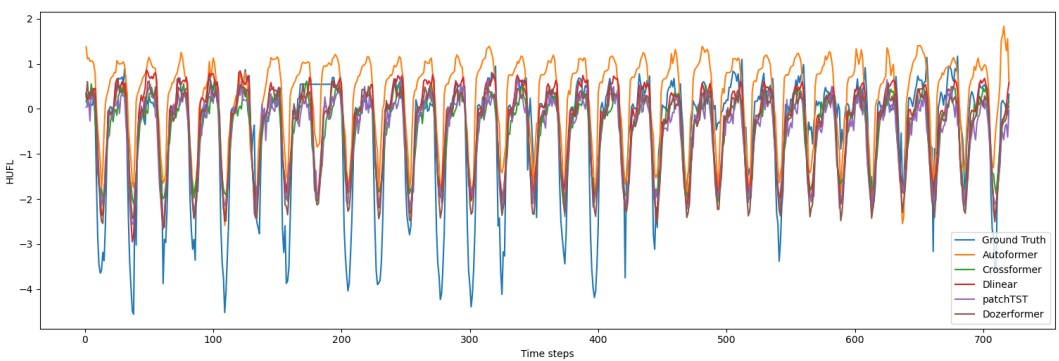

Figure 5: Visualizations of forecasting results on ETTh1 dataset at horizon 720.

### A.3.4 MORE ABLATION STUDIES

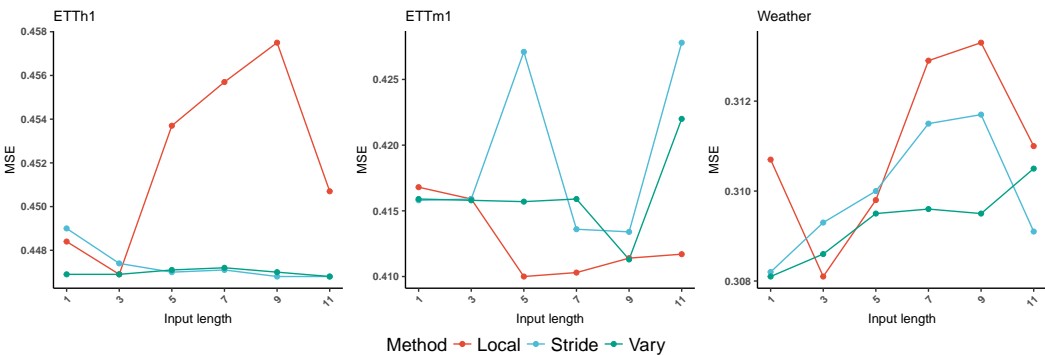

Figure 6: The forecasting results in terms of MSE for different Local, Stride, and Vary values at the horizon of 720.

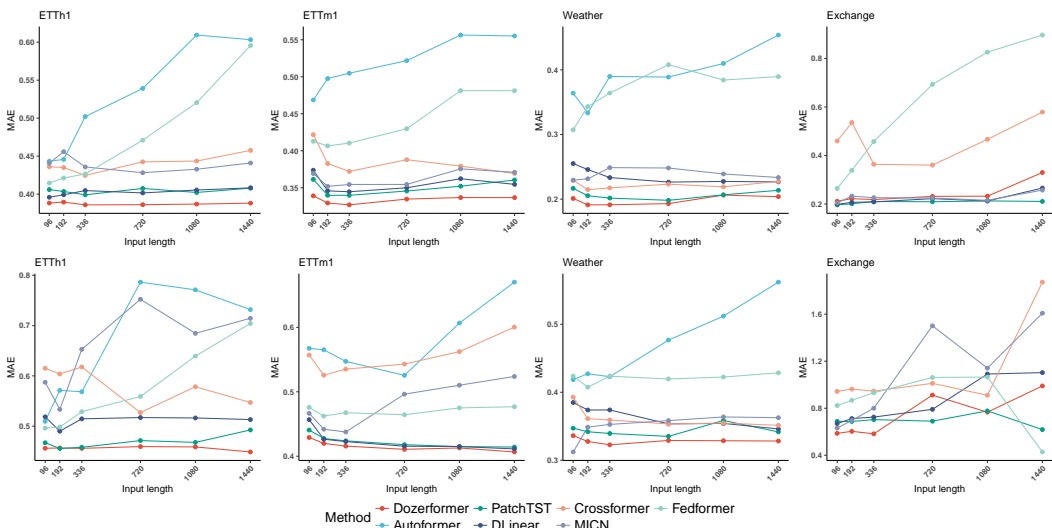

Figure 7: The forecasting results in terms of MAE for different look-back window sizes at horizons 96 and 720.

