# OpenReview forum: "Dozerformer: Sequence Adaptive Sparse Transformer for Multivariate Time Series Forecasting"
_ICLR.cc/2024/Conference — Submitted to ICLR 2024_

### Official Review · Reviewer_nz5f · 2023-10-25

**Soundness:** 2 fair
**Presentation:** 3 good
**Contribution:** 2 fair
**Rating:** 3
**Confidence:** 4

**Summary:**

To address the challenges of multivariate time series forecasting, this paper introduces 'Dozerformer,' an innovative approach that incorporates adaptive sparse attention mechanisms. Drawing insights from the analysis of seasonality and locality within the attention maps of Transformers, it introduces the concepts of 'local attention' and 'stride attention.' Moreover, recognizing that using the entire historical sequence may not always be beneficial for accurate forecasting, the paper introduces 'varying attention.' This feature adjusts the number of historical observations attended to based on the forecasting positions. As a result, Dozerformer demonstrates exceptional performance when compared to eight different baseline models across a range of forecasting tasks, effectively reducing attention-related computational costs.

**Strengths:**

They select an appropriate attention mechanisms to catch temporal dynamics in time series forecasting

**Weaknesses:**

**Problems in motivations**

1. In the second paragraph of the first page, the authors argue "predicting the value at horizon 1 using the entire historical sequence is suboptimal and inefficient.". This is the main motivation of vary attention in cross-attention modules. However, the reason for suboptimality is not provided in this paragraph. Why do we have to use varying-sequence attention?

2. The target task is multivariate time series forecasting. In this task, considering inter-feature connections is also important as well as temporal connections [1]. However, there is a discrepancy between the characteristics of the main task and your method. In other words, Dozerformer doesn't incorporate any parts to consider inter-feature dependencies. Can you explain the reason for this design or Do I understand wrong?

3. In the second paragraph of the first page, the authors argue "They (might include Transformers with full attention) also ignored the characteristics of MTS data, like locality and seasonality.". This is the reason for local and stride attention. However, when observing Figure 1 (a), Transformers with full attention already capture locality and seasonality automatically. At this point, why do we have to make constraints on self-attention?

4. For the motivation of local attention, the authors mention local properties in Figure 3(a) (it might be Figure 1(a)) in Section 3.2.1. However,   the tokens of the attention map in Figure 1(a) are observations. However, because your method includes patchifying, the tokens of yours are patches. The characteristics of patches cannot be directly explained by that of observations.

**Incomplete equations**

1. I recommend that the detailed formula of the dimension invariant embedding (DI Embed) should be included. I think many people might encounter DI Embed for the first time because [2] including DI Embed is not widely known. This makes readers easy to understand.

2. When decomposing time series into seasonal and trend parts, you might use average pooling and residual techniques. Although the same techniques are used in other papers, I recommend mentioning average pooling in your manuscript. Also, I'm curious about what kernel size is used for this average pooling.

3. For the Linear model to process $X_t$, can you include the formula of Linear model in the manuscript?

4. Eq. (2), (3), and (4) can be written more clearly, for example, with 'if else'.

5. In Eq. (4), I think something is wrong because when i = 1 and v = 3, $t-v+i-t<0$. Indices cannot be negative. Can you fix it?

**Insufficient experiments**

1. In Section 4.3, you just give us theoretical complexity. Can you give a cost comparison in an empirical way such as flops or wall-clock time?

2. In Table 3. it seems that local, stride, and vary attention modules are not quite helpful for forecasting in ETTh1 and ETTm1. Can you give more experimental results to prove the efficacy of the three modules?

[1] Zhang et al., Crossformer: Transformer Utilizing Cross-Dimension Dependency for Multivariate Time Series Forecasting, 2023, ICLR
[2] Zhang et al., Multi-scale transformer pyramid networks for multivariate time series forecasting

**Questions:**

See the weakness part.

---

> ### Author Response · Authors · 2023-11-19
> **Author's Response: Part I**
>
> We extend our gratitude to the reviewers for their meticulous evaluation and insightful comments. We have diligently addressed the questions as outlined below. If there are any additional concerns, we are more than willing to provide further clarification and resolution.
>
> **Problems in motivations**
>
>  >**1. In the second paragraph of the first page, the authors argue "predicting the value at horizon 1 using the entire historical sequence is suboptimal and inefficient.". This is the main motivation of vary attention in cross-attention modules. However, the reason for suboptimality is not provided in this paragraph. Why do we have to use varying-sequence attention?**
>
>  **Author's response:** In Figure 1(a), the heatmap illustrates the correlation among time steps. Notably, distant time steps exhibit a generally low correlation, while those in close proximity display higher correlation. Existing transformer-based methods, generate predictions for all future time steps in a single forward operation. This involves utilizing the entire historical sequence even for a forecasting horizon of 1, resulting in the inclusion of many historical time steps that are not correlated with the target. This practice not only hinders the model's ability to effectively learn useful patterns but also involves unnecessary computations. To address this, we introduced the Vary component, which adaptively increases the historical time steps used in cross-attention as the forecasting horizon grows. This adaptive approach avoids unnecessary computations from time steps not correlated with the targets, thereby enhancing the learning of temporal dependencies.
>
>  >**2. The target task is multivariate time series forecasting. In this task, considering inter-feature connections is also important as well as temporal connections [1]. However, there is a discrepancy between the characteristics of the main task and your method. In other words, Dozerformer doesn't incorporate any parts to consider inter-feature dependencies. Can you explain the reason for this design or Do I understand wrong?**
>
>  **Author's response:** Yes, we adhere to the channel-independent design introduced by patchTST. Extensive ablation studies conducted by patchTST demonstrated the effectiveness of channel independence in the context of MTS forecasting. While we experimented with inter-dependency learning for Dozerformer, the results indicated a degradation in performance.
>
> >**3. In the second paragraph of the first page, the authors argue "They (might include Transformers with full attention) also ignored the characteristics of MTS data, like locality and seasonality.". This is the reason for local and stride attention. However, when observing Figure 1 (a), Transformers with full attention already capture locality and seasonality automatically. At this point, why do we have to make constraints on self-attention?**
>
>  **Author's response:** The conventional full attention mechanism encompasses the entire historical sequence, resulting in the calculation of attentions between all time steps. However, this design presents two limitations: (1) Time steps that lack correlation with the forecasting targets are included in the attention calculation, leading to suboptimal model effectiveness and unnecessary computations. (2) The full attention mechanism is burdened by quadratic computation and memory usage. In response, we introduce the Dozer attention, which eliminates time steps lacking correlation with the targets in the attention mechanism. This removal allows the model to focus exclusively on learning data patterns from time steps correlated with the targets. Consequently, this sparse attention design not only alleviates constraints but also enhances the model's ability for temporal dependency learning and overall efficiency.
>
> >**4. For the motivation of local attention, the authors mention local properties in Figure 3(a) (it might be Figure 1(a)) in Section 3.2.1. However, the tokens of the attention map in Figure 1(a) are observations. However, because your method includes patchifying, the tokens of yours are patches. The characteristics of patches cannot be directly explained by that of observations.**
>
>  **Author's response:** While Figure 1(a) presents a heatmap at the individual time step level, evident seasonalities exist at multiple time step levels for the ETTh1 and ETTm1 datasets. Similarly, the Weather and Exchange-Rate datasets exhibit distinct locality at multiple time step levels, observable in the presence of clear yellow cubes in the heatmap, indicating higher correlation at those time steps. In our model, we segment the time series data along the time step dimension, postulating that these patches align with the highlighted yellow cubes in the heatmap.

---

> ### Author Response · Authors · 2023-11-19
> **Author's Response: Part II**
>
> **Incomplete equations**
> >**1. I recommend that the detailed formula of the dimension invariant embedding (DI Embed) should be included. I think many people might encounter DI Embed for the first time because [2] including DI Embed is not widely known. This makes readers easy to understand.**
>
>  **Author's response:** As requested, a detailed description of the DI embedding is provided in the appendix A.2.
>
> >**2. When decomposing time series into seasonal and trend parts, you might use average pooling and residual techniques. Although the same techniques are used in other papers, I recommend mentioning average pooling in your manuscript. Also, I'm curious about what kernel size is used for this average pooling.**
>
>  **Author's response:** Yes, we adhered to the decomposition procedure outlined in MICN. In the appendix, we have provided a description indicating the use of average pooling. The kernel sizes employed were 13 and 17, consistent with the MICN setting. Further details about the decomposition procedure can be found in Appendix A.2.
>
> >**3. For the Linear model to process $X_t$, can you include the formula of Linear model in the manuscript?**
>
>  **Author's response:** As requested, we have provided a detailed description of the Linear model in the Appendix A.2. We appreciate the reviewer for highlighting the need for a comprehensive explanation of these computation procedures.
>
> >**4. Eq. (2), (3), and (4) can be written more clearly, for example, with 'if else'.**
>
>  **Author's response:** As requested, we have provided a detailed description of the Linear model in the Appendix A.2. We appreciate the reviewer for highlighting the need for a comprehensive explanation of these computation procedures.
>
> >**5. In Eq. (4), I think something is wrong because when i = 1 and v = 3, $t-v+i-t<0$. Indices cannot be negative. Can you fix it?**
>
>  **Author's response:** $t$ represents the current time, and the index of historical time steps starts from $t-1$. The historical time steps utilized by the Vary component gradually increase as the forecasting horizon increases but stop at the maximum input length. We have included a sentence in the manuscript to explicitly state that all the time steps utilized by the Vary component fall within the historical window.
>
> **Insufficient experiments**
>
> >**1. In Section 4.3, you just give us theoretical complexity. Can you give a cost comparison in an empirical way such as flops or wall-clock time?**
>
>  **Author's response:** As requested, we conducted experiments to compare the FLOPs of Dozerformer with baseline methods. Additionally, we evaluated efficiency in terms of Params (total number of learnable parameters) and Memory (maximum GPU memory consumption). The results are presented in Table 7 in the Appendix. Dlinear stands out as the only method more efficient than Dozerformer, thanks to its simple linear layer-only architecture. However, the proposed Dozerformer exhibits significantly greater efficiency than both transformer-based and CNN-based methods. For instance, its FLOPs value is only 5.8% of that of patchTST.
>
> >**2. In Table 3. it seems that local, stride, and vary attention modules are not quite helpful for forecasting in ETTh1 and ETTm1. Can you give more experimental results to prove the efficacy of the three modules?**
>
>  **Author's response:** In Table 4, we presented accuracy and efficiency metrics. The lower query-key pairs reduction rates for ETTh1 and ETTm1 compared to the other two datasets are due to specific hyperparameter settings. A larger stride value, smaller local window size, and smaller vary starting size result in a more sparse attention matrix, thus enhancing efficiency. Given the clear daily seasonality in the ETTh1 and ETTm1 datasets, we opted for a less sparse attention matrix.

---

> ### Comment · Reviewer_nz5f · 2023-11-20
> **Thank you**
>
> Thank you for your effort to resolve my concerns.
>
> However, some answers are not enough to fully resolve them.
>
> **Problems in Motivation**
>
> **Q1**. I think Figure 1(a) is not a good example for your argument that  "predicting the value at horizon 1 using the entire historical sequence is suboptimal.". For example, in ETTm1, the 0-th observation is quite highly related to the 100-th one. I think that it is quite controversial to use Figure 1(a) as the reason for your argument.
>
> **Q2** Can you provide the experimental results for this experiment?
>
> **Q3** What is the detailed reason for this argument that "Time steps that lack correlation with the forecasting targets are included in the attention calculation, leading to suboptimal model effectiveness and unnecessary computations.".
>
> **Q4** The block size in each attention map, is different. However, patchfying involves segmenting time series into the same length of patches. Therefore, your answer is insufficient for my questions.
>
> **Incomplete equations**
> Can you highlight the changes in different colors to recognize them?
>
> **Insufficient experiments**
>
> **Q2** Can you provide more experimental results in other dataset settings?

---

> > ### Author Response · Authors · 2023-11-22
> > **Author's Second Response: Part I**
> >
> > We greatly appreciate your prompt response. We have thoroughly addressed the questions as outlined below. If there are any additional concerns, we are more than willing to provide further clarification and resolution.
> >
> > **Problems in motivations**
> >
> >  >**1. I think Figure 1(a) is not a good example for your argument that "predicting the value at horizon 1 using the entire historical sequence is suboptimal.". For example, in ETTm1, the 0-th observation is quite highly related to the 100-th one. I think that it is quite controversial to use Figure 1(a) as the reason for your argument.**
> >
> >  **Author's response:** In Figure 1(a), it is evident that ETTh1 and ETTm1 exhibit distinct seasonal patterns. ETTm1, sampled every 15 minutes, displays a pronounced daily seasonality with 96 time steps. The Stride component in the Dozer attention is designed to capture such seasonality. Importantly, this doesn't conflict with the Vary component, as these three components are combined using an 'or' relationship to generate the final sparse attention matrix. With an increasing forecasting horizon, a longer historical sequence ensures that the model can capture temporal dependencies over a larger receptive field.
> >
> >  >**2. Can you provide the experimental results for this experiment?**
> >
> >  **Author's response:** We present the ablation study of Dozerformer and Dozerformer with inter-dependency learning as follows.
> > |  Models  |     | Dozerformer |         | Channel_dependent |         |
> > |:--------:|:---:|:-----------:|:-------:|:-----------------:|:-------:|
> > |  Metrics |     |     MSE     |   MAE   |        MSE        |   MAE   |
> > |   ETTh1  |  96 | 0.3625      | 0.3861  | 0.3812            | 0.3986  |
> > |          | 192 | 0.4056      | 0.4128  | 0.4065            | 0.4203  |
> > |          | 336 | 0.4322      | 0.4288  | 0.4348            | 0.4434  |
> > |          | 720 | 0.447       | 0.4594  | 0.5445            | 0.5089  |
> > |   ETTm1  |  96 | 0.2921      | 0.3347  | 0.3069            | 0.3462  |
> > |          | 192 | 0.329       | 0.3583  | 0.3463            | 0.3662  |
> > |          | 336 | 0.3593      | 0.3765  | 0.3765            | 0.3885  |
> > |          | 720 | 0.4159      | 0.4107  | 0.4262            | 0.4153  |
> > |  Weather |  96 | 0.1473      | 0.1931  | 0.1484            | 0.1939  |
> > |          | 192 | 0.1896      | 0.2326  | 0.1912            | 0.2324  |
> > |          | 336 | 0.2375      | 0.2726  | 0.2373            | 0.2721  |
> > |          | 720 | 0.3082      | 0.3227  | 0.3152            | 0.3294  |
> > | Exchange |  96 | 0.09276     | 0.2136  | 0.08849           | 0.2091  |
> > |          | 192 | 0.1667      | 0.2919  | 0.1905            | 0.3096  |
> > |          | 336 | 0.2664      | 0.3799  | 0.3709            | 0.4369  |
> > |          | 720 | 0.5769      | 0.5899  | 1.064             | 0.7863  |
> > |          | Mean| 0.31431     | 0.35398 | 0.36431           | 0.37857 |
> >
> >  This table demonstrates that the channel-independent version of Dozerformer performed significantly better than the channel-dependent version, achieving the best results in 27 out of 32 cases. It also has a lower average MSE and MAE than the channel-dependent version. This result is consistent with the ablation study conducted by patchTST and presented in Table 7 of their paper. Consequently, we have adopted the channel-independent design.
> >
> > >**3. What is the detailed reason for this argument that "Time steps that lack correlation with the forecasting targets are included in the attention calculation, leading to suboptimal model effectiveness and unnecessary computations.**
> >
> >  **Author's response:** The process of using a machine learning method to forecast time series data involves deriving inferences from $I$ time steps of historical records $X$ to predict $O$ future values $Y$. It is evident that historical records $X$ correlated with targets $Y$ enable the model to learn such inferences. Conversely, if historical records $X$ lack correlation with targets $Y$, a gap will exist between the learned model and the characteristics of targets $Y". The experimental results also support this observation. We conducted an ablation study by replacing the sparse Dozer attention with the canonical full attention in the Dozerformer framework. The critical difference between them is that the Dozer attention excludes uncorrelated time steps in the attention calculation. The quantitative results of MSE and MAE are presented in Table 4. The Dozer attention achieved the best results in 22 out of 32 cases, outperforming the canonical full attention, which achieved the best results in only 9 cases. Therefore, we conclude that including time steps lacking correlation with forecasting targets degrades the model's performance in both accuracy and efficiency.

---

> > > ### Author Response · Authors · 2023-11-22
> > > **Author's Second Response: Part II**
> > >
> > > >**4. The block size in each attention map, is different. However, patchfying involves segmenting time series into the same length of patches. Therefore, your answer is insufficient for my questions.**
> > >
> > >  **Author's response:** The variation in block size in each heatmap arises from displaying the correlation of variables from four distinct datasets, each possessing unique characteristics. The Patch operation serves to divide the time series into patches or subseries based on the specific attributes of the dataset under consideration. For instance, in the case of the ETTh1 dataset, the patch size is set to 24 to capture temporal dependencies at a 24-hour resolution. In contrast, for the ETTm1 dataset, the patch size is 48, representing a resolution of 12 hours. It is crucial to note that the distribution of time series data evolves over time, leading to variations in the heatmap. However, the patch size signifies the framework's objective of modeling temporal dependencies at a specific scale. In all the methods [1][2] incorporating the Patch operation, the patch size remains fixed for a particular dataset.
> > >
> > > **Incomplete equations**
> > > >**1. I recommend that the detailed formula of the dimension invariant embedding (DI Embed) should be included. I think many people might encounter DI Embed for the first time because [2] including DI Embed is not widely known. This makes readers easy to understand.**
> > >
> > >  **Author's response:** Yes. We described the DI embed in detail in the appendix A.2 as follows:
> > > The DI embedding is utilized to embed the raw MTS into feature maps and partition them into patches.
> > > $$
> > > X_{emb} =Conv\left(X_{s}\right)
> > > $$
> > > $$
> > > X_{pat}=Patch\left(X_{emb},X_{\mathbf{0}}, p\right)
> > > $$
> > > where the $X_{s} \in \mathbb{R}^{1\times I\times D}$ is the seasonal components of input, $X_{emb} \in \mathbb{R}^{c\times I\times D}$ represents $c$ feature maps embedded by a convolutional layer with kernel size of $3 \times 1$.
> > > The $Patch$ procedure divides the time series into $N_I = \lceil I/p \rceil$ non-overlapping patches of size $p$, yielding $X_{pat} \in \mathbb{R}^{c \times N_I \times p \times D}$. The $X_{\mathbf{0}}$ is zero-padding when the input sequence length $I$ is not divisible by the patch size $p$. The Dozerformer adheres to a channel-independent design, where each variable is individually inputted into the transformer. Consequently, we combined the feature map dimension and the patch size dimension to form the transformer's input $X_{pat}^{d} \in \mathbb{R}^{N_I \times \left(p\times c\right)}$.
> > >
> > > >**2. When decomposing time series into seasonal and trend parts, you might use average pooling and residual techniques. Although the same techniques are used in other papers, I recommend mentioning average pooling in your manuscript. Also, I'm curious about what kernel size is used for this average pooling.**
> > >
> > >  **Author's response:** The procedure of the decomposition are described in detail in the Appendix A.2 and as follows:
> > > Given MTS input $X \in \mathbb{R}^{I\times D}$, the first step is to decompose it into seasonal and trend component as follows:
> > > $$
> > > X_{t}=mean(\sum_{i=1}^{n}AvgPool_i(Padding(X)))
> > > $$
> > > $$
> > > X_{s}= X - X_{t}
> > > $$
> > > Where $X_{s}\in \mathbb{R}^{I\times D}$ and  $X_{t}\in \mathbb{R}^{I\times D}$ are seasonal and trend-cyclical components, respectively.
> > >
> > >
> > > >**3. For the Linear model to process $X_t, can you include the formula of Linear model in the manuscript?**
> > >
> > >  **Author's response:** Yes. We described the Linear model in detail in the Appendix A.2 as follows:
> > > A simple linear layer is utilized to model the trend component and generate trend component predictions.
> > > $$
> > > X^{pred}\_{t} = Linear(X\_t)
> > > $$
> > > where $X^{pred}_{t} \in \mathbb{R}^{O\times D}$ is the prediction for trend component. The $Linear$ operation directly generates $O$ future trend values by projecting from $I$ historical trend values for each variable in the MTS data.
> > >
> > >
> > > >**4.	Eq. (2), (3), and (4) can be written more clearly, for example, with 'if else'.**
> > >
> > >  **Author's response:** We agree with the reviewer. We changed “for” to “if else” in those equations.
> > >
> > >
> > > **Insufficient experiments**
> > >
> > > >**1.	Can you provide more experimental results in other dataset settings**
> > >
> > >  **Author's response:** Certainly, Figure 6 in Appendix A.3.4 illustrates the MSE results of Dozerformer with varying local, stride, and vary sizes from 1 to 11. Generally, the optimal values for these parameters differ across datasets. Notably, a sparse attention matrix consistently outperforms a dense one, demonstrating its efficiency.
> > >
> > >
> > >
> > > References:
> > >
> > > [1] Yuqi Nie, Nam H. Nguyen, Phanwadee Sinthong, and Jayant Kalagnanam. A time series is worth 64 words: Long-term forecasting with transformers. In International Conference on Learning Representations, 2023.
> > >
> > > [2] Yunhao Zhang and Junchi Yan. Crossformer: Transformer utilizing cross-dimension dependency for multivariate time series forecasting. In International Conference on Learning Representations, 2023.

---

> ### Comment · Reviewer_nz5f · 2023-11-23
> **Thank you**
>
> I appreciate your effort to resolve my concerns.
>
> However, I want more concrete observations or experimental results to justify your design or arguments. Without concrete results or other related papers, only some weak intuition is not enough. Therefore, I uphold my scores as before.

---

### Official Review · Reviewer_Pv5r · 2023-10-28

**Soundness:** 3 good
**Presentation:** 3 good
**Contribution:** 3 good
**Rating:** 6
**Confidence:** 2

**Summary:**

This work introduces Dozerformer that has the Dozer attention mechanism at its core. The proposed approach addresses the quadratic time complexity and the focus on full history for applying attention in traditional transformers. The Dozer attention allows for attending to a local window, and also selected past keys. It also presents a stride mechanism to that allows attending at predefined intervals. Ablation studies and empirical explorations are made available to support the proposed approach.

**Strengths:**

Originality :

The work is somewhat original as it combines multiple attention formulations (local, stride, vary) to produce a joint efficient formulation.

Quality :

This work creates a well structured framework and proposes an efficient attention mechanism for MTS.  The proposed approach is supported by ablation studies and explorations on several datasets. However, there are some questions that come up.

Clarity :

This work is somewhat clear although there are some parts (such as those mentioned under Questions) that could be made more clear.


Significance:

Efficient attention mechanisms are important not just conceptually for forecasting but also for practical training of the model. To this end, the proposed work presents a step forward.

**Weaknesses:**

Although the work is well structured and conceptually the proposed formulations can be beneficial, it can benefit by providing additional clarity and evidence through different downstream tasks (as indicated in the Questions section)

**Questions:**

Figure 1b-stride is a bit unclear. Why is it that t+1 and t+O are attending to different number of past steps. I am assuming that each of figure in 1b refer to the corresponding specific component (such as 'full', 'local' etc.). Furthermore, what is the stride value in 1b-stride.


Section 3.1 is a bit unclear. How are the outputs of 1x1 conv and the linear layers combined. I assume that the linear layer produce an output $\in \mathbb{R}^{I \times D_1}$ which is then projected to  $X_{pred}$ after combining with the output of 1x1 conv. However it is not clear how the I rows of $X_{t}$ matrix get converted to O rows so that they can be added to the output of 1x1 conv.


The explorations are performed on datasets that contain signals that perhaps do not change at a very fast rate rate. It would be good to have experiments on fast changing data. The effectiveness of local , stride and vary can be more clear on such datasets.


It is also not clear about the practical significance of the difference in forecasted numbers from the different model. Ideally, the forecasted numbers would be used in another appropriate downstream task to show how the different forecasting mechanisms perform. MSE/MAE numbers are perhaps not enough. For example, in Figure 4/5 , several forecasting approaches produce plots that are very close to each other. Therefore, they might perform equivalently when the forecasted data is used for downstream task.


As efficiency is one of the focus areas of this work, if possible, it would be good to have actual computation numbers (such as memory and wall-clock time) for the proposed approach. A comparison of such numbers from other baselines (if available) would also be useful.

---

> ### Author Response · Authors · 2023-11-19
> **Author's Response: Part I**
>
> We appreciate the thorough review and insightful comments from the reviewers. Should there be any additional concerns, we are more than willing to provide further clarification and address them.
>
>  >**1. Figure 1b-stride is a bit unclear. Why is it that t+1 and t+O are attending to different number of past steps. I am assuming that each of figure in 1b refer to the corresponding specific component (such as 'full', 'local' etc.). Furthermore, what is the stride value in 1b-stride.**
>
>  **Author's response:** Yes, each subfigure in Figure 1b represents a distinct component. Specifically, the subfigure titled "Stride" denotes the Stride component within the Dozer attention mechanism. The Stride component selectively attends queries to keys located at fixed intervals.
> In Figure 1b, the stride value is set to 3. The variance in historical time steps utilized by *t+1* and *t+O* stems from the fact that the historical time steps '*I*' in the figure amount to 8, with indexes ranging from *t-8* to *t-1*. Consequently, the selected time steps could be 2 or 3 for a sample rate of 3, determined by the starting index. To illustrate, when the forecasting target is *t+1*, the historical time steps selected for a stride of 3 are *t-3* and *t-6*. For a forecasting target at *t+2*, the historical time steps selected are *t-2*, *t-5*, and *t-8*.
>
>  >**2. Section 3.1 is a bit unclear. How are the outputs of 1x1 conv and the linear layers combined. I assume that the linear layer produce an output $\in \mathbb{R}^{I\times D_1} $ which is then projected to $X_{pred}$ after combining with the output of 1x1 conv. However it is not clear how the I rows of $X_t$ matrix get converted to O rows so that they can be added to the output of 1x1 conv.**
>
>  **Author's response:** The Linear model serves as the trend model, tasked with directly inferring predictions for O future time steps of the trend component from I historical records. The outputs’ dimension of both the Linear model and the 1x1 convolution is OxD, after which they undergo elementwise summation to produce the final forecasting values. We express gratitude to the reviewer for highlighting that certain computation procedures were omitted in the manuscript due to page limits. Consequently, we have included a step-by-step detailed computation description of the framework in Appendix A.2.
>
> >**3. The explorations are performed on datasets that contain signals that perhaps do not change at a very fast rate rate. It would be good to have experiments on fast changing data. The effectiveness of local , stride and vary can be more clear on such datasets.**
>
>  **Author's response:** The nine benchmark datasets employed in the experiments vary in sample rate from 10 minutes to 1 week. Some datasets exhibit rapid changes. For instance, Figure 4 and Figure 5 visualize the ETTh1 dataset, illustrating that its values fluctuate between -4 and 1 in less than 24 time steps. If additional datasets with faster-changing characteristics are available and accessible via a provided download link, we are open to incorporating them. We believe that the current selection of nine widely used benchmark datasets already effectively demonstrates the efficacy of Dozerformer in terms of both accuracy and efficiency.
>
> >**4. It is also not clear about the practical significance of the difference in forecasted numbers from the different model. Ideally, the forecasted numbers would be used in another appropriate downstream task to show how the different forecasting mechanisms perform. MSE/MAE numbers are perhaps not enough. For example, in Figure 4/5 , several forecasting approaches produce plots that are very close to each other. Therefore, they might perform equivalently when the forecasted data is used for downstream task.**
>
>  **Author's response:** I concur with the reviewer. While the improvements in MSE and MAE values are notable, their impact on downstream tasks may not be substantial. As depicted in Figure 4 and Figure 5, the forecasting results of all methods exhibit close proximity. I acknowledge that their performance might be similar for downstream tasks. This is an aspect often overlooked in the field. Notably, recent state-of-the-art papers [1][2][3] have enhanced quantitative metrics, yet their predictions, as visualized, do not exhibit significant differences. It raises a pertinent issue regarding the potential lack of discernible performance distinctions in downstream tasks. I agree that further investigation into this matter is warranted, representing a gap in the field, particularly concerning the application of deep learning models in downstream tasks and understanding how forecasting accuracy influences these tasks.

---

> ### Author Response · Authors · 2023-11-19
> **Author's Response: Part II**
>
> >**5. As efficiency is one of the focus areas of this work, if possible, it would be good to have actual computation numbers (such as memory and wall-clock time) for the proposed approach. A comparison of such numbers from other baselines (if available) would also be useful.**
>
>  **Author's response:** As requested, we present experimental results comparing the proposed Dozerformer with baseline methods, illustrating the parameters (total number of learnable parameters), FLOPs (floating-point operations), and memory (maximum GPU memory consumption) of all methods in Table 7 in the appendix. Dlinear stands out as the only method more efficient than Dozerformer, owing to its simple linear layer-only architecture. However, the proposed Dozerformer exhibits significantly greater efficiency than both transformer-based and CNN-based methods. For instance, its FLOPs value is only 5.8% of that of patchTST.
>
>
> References:
>
> [1] Yuqi Nie, Nam H. Nguyen, Phanwadee Sinthong, and Jayant Kalagnanam. A time series is worth 64 words: Long-term forecasting with transformers. In International Conference on Learning Representations, 2023.
>
> [2] Yunhao Zhang and Junchi Yan. Crossformer: Transformer utilizing cross-dimension dependency for multivariate time series forecasting. In International Conference on Learning Representations, 2023.
>
> [3] Haixu Wu, Jiehui Xu, Jianmin Wang, and Mingsheng Long. Autoformer: Decomposition transformers with Auto-Correlation for long-term series forecasting. In Advances in Neural Information Processing Systems, 2021.

---

> > ### Comment · Reviewer_Pv5r · 2023-11-22
> >
> > Thanks for the author response and clarifications

---

### Official Review · Reviewer_zXpM · 2023-10-30

**Soundness:** 3 good
**Presentation:** 2 fair
**Contribution:** 2 fair
**Rating:** 3
**Confidence:** 4

**Summary:**

Transformer based models of existing time series forecasting show quadratic time complexity. In addition, it shows the limitations of existing self-attention in making predictions using the entire input sequence. This paper shows excellent results in terms of performance and complexity by paying attention only to the historical data needed for prediction. To do this, this paper proposes dozer attention. dozer attention

It consists of three parts: Local, Stride, and Vary. Each component captures temporal dependencies from time-points deemed important.

Dozerformer is an adaptive sparse transformer framework, the core of which is sparse Dozer attention.

**Strengths:**

1. It is meaningful that motivation is clear and attention is defined using the inductive bias of multivariate time series data for self-attention.
2. Comparison of all recent state-of-the-art models.

**Weaknesses:**

1. It would be nice to have a structural comparison or analysis with Pyraformer, Informer, and PatchTST, which propose sparse attention. Furthermore, I would like to know which aspect of Dozerformer shows a better contribution.
2. Performance is not very good compared to PatchTST and Dlinear (LTSF-Linear). Is there any reason or basis for this?
3. Although the motivation is clear, I think this paper lacks a point of differentiation from many existing similar studies.

**Questions:**

1. It would be nice to have a structural comparison or analysis with Pyraformer and Informer, which proposed sparse attention. Furthermore, I would like to know which aspect of Dozerformer shows better contribution.
- minor typos
    - In 3.2.1 Figure 3 (a) —> Figure 3

---

> ### Author Response · Authors · 2023-11-19
> **Author's Response**
>
> We express our gratitude to the reviewers for their meticulous evaluation and insightful comments. We have diligently addressed the questions as outlined below. If there are any additional concerns, we are more than willing to provide further clarification and resolution.
>  >**1. It would be nice to have a structural comparison or analysis with Pyraformer and Informer, which proposed sparse attention. Furthermore, I would like to know which aspect of Dozerformer shows better contribution.**
>
>  **Author's response:** The Informer introduced a ProbSparse attention mechanism, selecting only the top-u queries based on query sparsity measurements. In contrast, the Dozer attention, informed by dataset characteristics (locality, seasonality, global correlations), avoids additional computations when choosing a subset of keys. As a comprehensive framework, the Dozerformer significantly outperformed the Informer in terms of accuracy, reducing MSE and MAE by 66.2% and 51.9%, respectively.
> The Pyraformer proposed a pyramidal attention module that calculates attention within a C-ary tree. This mechanism selectively attends each query to a limited set of keys, categorized into three types: adjacent keys at the same level (scale) in the graph, keys of parent nodes connected to the query, and keys of child nodes connected to the query. Consequently, the Pyraformer models temporal dependencies at multiple resolutions. However, its notable drawback is the absence of modeling global temporal dependencies at a specific resolution; instead, it attempts to capture local temporal dependencies at a coarse scale. Consequently, its performance is less accurate compared to our proposed Dozerformer, which reduced MSE and MAE values by 64.8% and 51.6%, respectively.
> We present a computational comparison of all methods in Table 7 in the manuscript's appendix. Notably, the Pyraformer and Informer stand out as two of the most efficient transformer-based methods. However, the Dozerformer achieved lower FLOPs than both methods, amounting to 1/10 of the Pyraformer and 1/15 of the Informer.
> In conclusion, while both the Informer and Pyraformer have proposed sparse attention mechanisms, our proposed Dozerformer outperforms them significantly in terms of accuracy. Additionally, the Dozerformer exhibits greater efficiency, as evidenced by the FLOPs comparison
>
>
>  >**2. Performance is not very good compared to PatchTST and Dlinear (LTSF-Linear). Is there any reason or basis for this?**
>
>  **Author's response:** As outlined in the paper, the Dozerformer demonstrated a 0.4% reduction compared to PatchTST and an 8.3% reduction compared to Dlinear in terms of Mean Squared Error (MSE). We consider the accuracy improvements to be substantial compared to Dlinear. Notably, PatchTST employs the canonical full attention mechanism, while our method incorporates the proposed Dozer Attention, where each query selectively attends to a subset of keys. Consequently, this approach reduces computational complexity to linear. While the Dozerformer's accuracy improvement over PatchTST may be modest, it achieves a significant enhancement in computation and memory efficiency. Consider ETTh1 with a forecasting horizon of 96 and a historical sequence length of 720 as an example. In this case, the Floating Point Operations Per Second (FLOPS) for Dozerformer amount to 14.91, which is markedly lower than patchTST, which registers at 256.39. Additionally, the parameter count for Dozerformer is 1.61 million, substantially less than patchTST, which has 10.73 million parameters.
>
> >**3. Although the motivation is clear, I think this paper lacks a point of differentiation from many existing similar studies.**
>
>  **Author's response:** The primary contribution of this paper lies in the introduction of the Dozer attention mechanism, comprising three components: local, stride, and vary. The Vary component is particularly novel compared to existing forecasting methods, as it adaptively adjusts the utilization of historical time steps with an increasing forecasting horizon, illustrated in Figure 1 (b). Unlike other transformer-based methods that generate predictions for all future time steps in a single forward operation, employing the same number of historical time steps, the Dozer attention recognizes that time steps further from the forecasting target are less correlated. Thus, the optimal historical sequence length varies with different forecasting horizons, especially for datasets lacking periodicity, as depicted in Figure 1 (a). To the best of our knowledge, the proposed Dozer attention is the only time series forecasting method that dynamically adapts to varying historical records.

---

### Official Review · Reviewer_xCiy · 2023-11-01

**Soundness:** 3 good
**Presentation:** 3 good
**Contribution:** 3 good
**Rating:** 6
**Confidence:** 4

**Summary:**

This article proposes some new attention mechanisms for Transformers in the context of Multivariate Time Series (MTV) forecasting. The main idea is to segregate local, seasonal, and global temporal dependencies and capture them through independent/corresponding attention portfolios. They call their approach sequence-adaptive Dozer Attention, and it comprises three sparse attention components (a) Local (to attend local relationship), (b) Stride (to attend intervals), and (c) Vary (to selectively attend to keys of historical sequence).

The authors have compared their method with some state-of-the-art methods through nine benchmark datasets. Results look promising;  also, the proposed method shows improved complexity analysis results for certain settings.

**Strengths:**

The ideas of signal locality, seasonality, and global temporal dependency are quite standard in time series processing. Connecting these notions to corresponding attention mechanisms, especially in the context of Multivariate Time Series (MTV) forecasting, look to be quite new, and one of the major contributions of this work.

Overall, the paper is well organized, clearly written, and it is easy to follow.  The proposed method has been tested on a number of benchmarks, and reported results are found to be promising. In addition, the computational complexity analysis and the attention mechanism ablation study have added some extra points to the work.  The paper exhibits some potential significance in the field of Multivariate Time Series forecasting.

**Weaknesses:**

Most of the results are reported using MAE and MSE as evaluation matrices. I would suggest adding Mean Absolute Scaled Error (MASE) in to the evaluation metric mix, as this an important metric for many time series prediction problems.

Reported results over different benchmarks look promising; however, none of the experiments include any significance tests. So, it is hard to evaluate if the results are statistically significant or not. I would also suggest running some statistical significance tests when comparing results.

We can get rid of the third digit after the decimal point in Table 1(page 7); this may improve the readability of the content.

**Questions:**

I have a few suggestions which may improve the quality of the paper:

(a) To add Mean Absolute Scaled Error (MASE) in to the evaluation metric mix, (b) To perform some statistical significance tests, so we can be sure the results are not random, (c) We can get rid of the third digit after the decimal point in Table 1 (page 7); this should improve the readability of the content.

One question: Did you only try grid search as your HP optimization technique (section 4.1)?

---

> ### Author Response · Authors · 2023-11-19
> **Author's Response: Incorporated the requested MASE metric and conducted the statistical test.**
>
> We thank the reviewer’s effort to read and review this paper. We respond to questions as follows.
>
> > **1. To add Mean Absolute Scaled Error (MASE) into the evaluation metric mix,**
>
> **Author's response:** We followed the request and incorporated Mean Absolute Scaled Error (MASE) presented in Table 6 in the manuscript's appendix. The average MASE for all cases is 0.604, better than the rest baseline methods.
>
> > **2. To perform some statistical significance tests, so we can be sure the results are not random,**
>
> **Author's response:** The main results presented in Table 1 represent the mean values obtained from six runs with distinct seeds. In compliance with the request, we computed the standard deviation (STD) across these six runs. Notably, for the ILI dataset, the results exhibit a comparatively large average STD in MSE and MAE, amounting to 0.164 and 0.053, respectively, across four forecasting horizons. The average range for these four horizons is 0.419 and 0.137. This discrepancy is attributed to the substantially smaller forecasting horizon in comparison to the other datasets. Nevertheless, the results consistently outperform baseline methods, except for patchTST, even for seeds generating higher MSE and MAE. Across the remaining eight datasets, the average STD for MSE and MAE stands at 0.002 and 0.001, with an average range of 0.007 and 0.003. This underscores the robustness of the results against randomness and underscores the consistently superior performance of the proposed Dozerformer.
>
> > **3. We can get rid of the third digit after the decimal point in Table 1 (page 7); this should improve the readability of the content.**
>
> **Author's response:** We acknowledge that Table 1 appears somewhat congested. To be consistent with baseline methods, we maintained three digits after the decimal point.
>
> > **4. Did you only try grid search as your HP optimization technique (section 4.1)?**
>
> **Author's response:** Yes, I only employed the grid search technique to select hyperparameters. For those parameters not chosen through grid search, we followed the settings of recent SOTA papers.

---

### Meta-Review · Area_Chair_1SaE · 2023-12-06

**Metareview:**

The paper proposed Dozerformer - an innovative approach that incorporates adaptive sparse attention mechanisms to address the challenges of multivariate time series forecasting. The reviewers appreciate the authors’ effort to resolve the reviewers’ concerns. However, they want more concrete observations or experimental results to justify the authors’ design or arguments. Without concrete results or other related papers, only some weak intuition is not enough. Therefore, the reviewers decide not to support the paper.

**Justification For Why Not Higher Score:**

Most of the reviewers do not support the paper.

**Justification For Why Not Lower Score:**

N/A

---

### Decision · Program_Chairs · 2024-01-16

Reject